



# The Effects of Spatial and Temporal Resolution of Gridded Meteorological Forcing on Watershed Hydrological Responses

Pin Shuai[1], Xingyuan Chen[1], Utkarsh Mital[2], Ethan T. Coon[3], and Dipankar Dwivedi[2]

[1]Pacific Northwest National Laboratory, Richland, WA 99352
[2]Lawrence Berkeley National Laboratory, Berkeley, CA 94720
[3]Climate Change Science Institute & Environmental Sciences Division, Oak Ridge National Laboratory, Oak Ridge, TN, 37830

**Correspondence:** Pin Shuai (pin.shuai@pnnl.gov)

**Abstract.** Meteorological forcing plays a critical role in accurately simulating the watershed hydrological cycle. With the advancement of high-performance computing and the development of integrated watershed models, simulating the watershed hydrological cycle at high temporal (hourly to daily) and spatial resolution (10s of meters) has become efficient and computationally affordable. These hyperresolution watershed models require high resolution of meteorological forcing as model input to ensure the fidelity and accuracy of simulated responses. In this study, we utilized the Advanced Terrestrial Simulator (ATS),
an integrated watershed model, to simulate surface and subsurface flow and land surface processes using unstructured meshes at the Coal Creek Watershed near Crested Butte (Colorado). We compared simulated watershed hydrologic responses including streamflow, and distributed variables such as evapotranspiration, snow water equivalent (SWE), and groundwater table driven by three publicly available, gridded meteorological forcing (GMF) – Daily Surface Weather and Climatological Summaries (Daymet), Parameter-elevation Regressions on Independent Slopes Model (PRISM), and North American Land Data Assim-
ilation System (NLDAS). By comparing various spatial resolutions (ranging from 400 m to 4 km) of PRISM, the simulated streamflow only becomes marginally worse when spatial resolution of meteorological forcing is coarsened to 4 km (or 30% of the watershed area). However, the 4 km resolution has much worse performance than finer resolution in spatially distributed variables such as SWE. By comparing models forced by different temporal resolutions of NLDAS (hourly to daily), GMF
in sub-daily resolution preserves the dynamic watershed responses (e.g., diurnal fluctuation of streamflow) that are absent in results forced by daily resolution. Conversely, the simulated streamflow shows better performance using daily resolution compared to that using sub-daily resolution. Our findings suggest that the choice of GMF and its spatiotemporal resolution depends on the quantity of interest and its spatial and temporal scale, which may have important implications on model calibration and watershed management decisions.

# 1 Introduction

The accuracy of meteorological forcings such as precipitation plays a crucial role in simulating watershed hydrological cycle. With the advancement of high-performance computing and the development of integrated hydrologic models (e.g., Amanzi-Advanced Terrestrial Simulator (ATS) (Coon et al., 2019), ParFlow (Kollet and Maxwell, 2006), HydroGeoSphere (Aquanty,





2015), and Weather Research and Forecasting Hydrological model (WRF-Hydro) (Gochis et al., 2018)), simulating watershed
hydrological cycle at high temporal and spatial resolution has become possible (Wood et al., 2011). These models often require
gridded meteorological forcing (GMF), which is typically fused from various sources, including ground-based gages, radar,
satellite remote sensing, as well as regional and global climate models. Due to different interpolation methods and data sources,
GMF is available at different spatial and temporal resolutions and contains considerable uncertainties (Schreiner-McGraw and
Ajami, 2020).

Recently, GMF products, notably Daily Surface Weather and Climatological Summaries (Daymet) (Thornton et al., 1997, 2021),
Parameter-elevation Regressions on Independent Slopes Model (PRISM) (Daly et al., 2008) and North American Land Data
Assimilation System (NLDAS) (Mitchell, 2004; Xia et al., 2012)), have become popular for hydrologic modeling within the
CONUS owing to their temporally and spatially complete coverage and relatively high spatiotemporal resolution. Past studies
have compared and evaluated the performance of GMF against weather stations (Behnke et al., 2016; Daly et al., 2008; Muche
et al., 2020). Daly et al. (2008) presented a detailed comparison between PRISM and Daymet and found that, for the products
available in 2008, PRISM outperforms Daymet, especially in mountainous and coastal areas of the western U.S. Behnke et al.
(2016) compared eight widely used meteorological forcing datasets including Daymet, PRISM, and NLDAS against Global
Historical Climatology Network-Daily (GHCN-D) stations across the nous US. They found that different interpolation meth-
ods affected the accuracy of downscaled meteorological data and care should be taken when selecting meteorological forcing
for a given region. In a similar study, Muche et al. (2020) compared four GMFs (i.e., Daymet, PRISM, NLDAS, and Global
Land Data Assimilation System (GLDAS)) as precipitation data sources, and evaluated the precipitation estimates at GHCN-D
stations within the Delaware Watershed at Perry Lake in eastern Kansas. They showed that precipitation from Daymet and
PRISM were more closely matched with precipitation collected at GHCN-D than that from NLDAS and GLDAS.

Understanding the bias and fidelity of each meteorological forcing and the effects of meteorological forcing spatiotemporal
resolution on simulated watershed responses is important for accurate simulations of watershed processes. Previous studies
have evaluated the impact of different meteorological forcing on model simulated surface runoff and streamflow (Muche et al.,
2020; Behnke et al., 2016; Gao et al., 2017; Elsner et al., 2014). Using the Soil and Water Assessment Tool (SWAT), Muche
et al. (2020) evaluated model performance on simulated streamflow against observation under different meteorological forc-
ing. They found that the simulated streamflows yielded a higher correlation when driven by PRISM and Daymet than those by
NLDAS and GLDAS. Eum et al. (2014) evaluated hydrologic responses using the Variable Infiltration Capacity (VIC) model
forced by three gridded meteorological datasets available in Canada. They found notable differences in simulated surface runoff
during the snow-melt period, but not so much during the snowfall period. However, these studies mostly focused on meteoro-
logical forcing effects on surface runoff and ignored other relevant hydrological processes (e.g., snowmelt, evapotranspiration
(ET), and subsurface flow). In addition, these studies used either semi-distributed models (e.g., SWAT) or coarse regional-scale
land surface models (e.g., VIC), which do not fully utilize the GMF at their finest resolutions.

Compared to semi-distributed models, fully-distributed, integrated hydrologic models are favorable in simulating watershed
hydrologic responses to changes in climate forcing as they can preserve the spatial heterogeneity of inputs from GMF and
provide a spatially distributed representation of both surface and subsurface flow processes. Recently, Maina et al. (2020)





used ParFlow-Community Land Model (CLM), a fully distributed, processes-based watershed model, to study the effect of
spatial resolution of meteorological forcing (0.5 to 40.5 km) generated from the WRF model on spatially resolved hydrologic
responses, including snow water equivalent (SWE), ET, infiltration, surface ponded depth and groundwater table. Using the
Cosumnes Watershed as a testbed, they found that most hydrologic variables were seasonally and spatially dependent on the
different spatial resolutions of the meteorological forcing. Although climate models such as WRF provide alternative GMF at
any given spatiotemporal resolution, they require extensive expert knowledge in setting up and running the models and thus are
less popular compared to publicly available GMF (e.g., Daymet, PRISM, and NLDAS). To our knowledge, few, if any, studies
have utilized the common GMFs to investigate the impact of spatial resolution of meteorological forcing on both watershed
cumulative variable (e.g., streamflow) and distributed variables (e.g., SWE, ET, and groundwater level).

    The temporal resolution of meteorological forcing input, especially precipitation, plays an important role in the timing of
runoff generation. It is particularly important for flood volume modeling (Ficchì et al., 2016), flood forecasting (Wetterhall
et al., 2011), and hydrodynamic modeling in urban catchment (Ochoa-Rodriguez et al., 2015; Bruni et al., 2015). The tem-
poral resolution of rainfall inputs has shown to affect the simulation of surface runoff more strongly than variations in spatial
resolution during storm events (Ochoa-Rodriguez et al., 2015). High temporal resolution is also important for studying wa-
tershed biogeochemical cycling since sub-daily meteorological forcing could induce diurnal snowmelt that produces regular
infiltration of cold, chemically distinct snow water into the soil which alters the soil temperature and chemical composition
of soil and ground water (Petrone et al., 2007; Woelber et al., 2018). Despite the importance of the temporal resolution of
input forcing, the impact of GMF temporal resolution on watershed hydrodynamics has largely been overlooked. For example,
a daily timestep is used routinely in watershed hydrologic modeling and the simulated daily streamflow is generally used to
compare with observed daily streamflow even though sub-daily streamflow is measured in most watersheds.

    The objective of this study is to intercompare three widely available GMFs (i.e., PRISM, Daymet, and NLDAS) and to
evaluate the impact of meteorological forcing spatial and temporal resolution on simulated watershed hydrologic responses
including streamflow, ET, SWE, soil moisture, ponded surface water depth, and groundwater table. We choose ATS as the
integrated watershed model to couple surface and subsurface flows with land surface processes (Coon et al., 2019). The model
can fully resolve the meteorological forcing on a much finer resolution ($<= 100\ m$) using unstructured triangular grids. We
seek to understand the impact of meteorological forcing by comparing model simulations with field observations including
GHCN-D stations, United States Geological Survey (USGS) stream gages, and remote sensing products. We aim to answer the
following questions: (1) How would different meteorological forcing in their native resolution impact the simulated streamflow,
distributed variables such as SWE and ET? (2) What is the effect of spatial resolution of GMF on simulated streamflow and
spatially distributed variables? (3) What is the effect of temporal resolution of GMF on simulated streamflow, and spatially dis-
tributed variables? and (4) Is spatial resolution more important than temporal resolution of the GMF for watershed simulations?
To address these questions, we perform different numerical experiments using ATS by forcing the model with various spatial
and temporal resolutions of GMFs. We choose a mountainous watershed due to its complex terrain and heterogeneous weather
conditions, which provides an ideal testbed for studying the impact of meteorological forcing spatiotemporal resolution on
watershed dynamic responses. The findings from this study are relevant for the use of GMF dataset on watershed hydrologic




simulations using fully distributed watershed models in mountainous watersheds. It also provides important implications on

watershed calibration using inverse modeling.

## 2 Methods

### 2.1 Study site

Our study site is located in the Coal Creek Watershed (Hydrologic Unit Code (HUC): 140200010204) with an area of $53.2\,km^2$ located within the larger East Taylor Watershed (HUC: 14020001) near Crested Butte, in southwestern Colorado (Figure 1).

The Coal Creek Watershed is a high alpine, snow-dominated catchment, characterized as warm summer, humid continental climate on the Koppen classification system (Koppen and Geiger, 1930). It receives ∼850 mm of precipitation annually, with ∼530 mm as snowfall. The primary land cover types are evergreen forest (62.6%) and shrub (20.5%). This watershed has strong variations in topography and land cover, which is representative of many headwater, mountainous watersheds in the western U.S.

### 2.2 ATS model setup

ATS is an integrated, distributed hydrologic code that solves the diffusion wave approximation of the St. Vernant equations for surface flow coupled to Richards equation for flow in variably saturated porous media in the subsurface (Coon et al., 2019, 2020). The Richards equation is described as:

$$\frac{\partial}{\partial x}(\phi s) + \triangledown \cdot \boldsymbol{q} = 0 \qquad (1)$$

with:

$$\boldsymbol{q} = -\frac{1}{\mu} k_r \kappa (\triangledown p + \rho g) \qquad (2)$$

where $\phi$ is the effective porosity [-], $s$ is the saturation [-], $\boldsymbol{q}$ is Darcy flux $[m/s]$, $\mu$ is the dynamic viscosity $[Pa \cdot s]$, $k_r$ is relative permeability [-], $\kappa$ is the saturated hydraulic permeability $[m^2]$, $p$ is water pressure [Pa], and $g$ is the gravitational constant $[m/s^2]$.

The diffusive wave approximation to overland flow is described as:

$$\frac{\partial h}{\partial t} + \triangledown \cdot (h\boldsymbol{v}) = Q_w + Q_{ss} \qquad (3)$$

with:

$$\boldsymbol{v} = -\frac{h^{2/3}}{n \cdot max(\epsilon, \sqrt{\triangledown z})} \triangledown(z + h) \qquad (4)$$

where $h$ is the depth of ponded water [m], $\boldsymbol{v}$ is the surface flow velocity [m/s], $Q_w$ is all external source/sink term [m/s],

$Q_{ss}$ is the exchange flux between surface and subsurface systems [m/s], $n$ is the Manning's coefficient $[s/m^{1/3}]$, $z$ is surface elevation [m], and $\epsilon$ is a small positve regularization to keep the equations non-singular in places with zero bed slope [m].

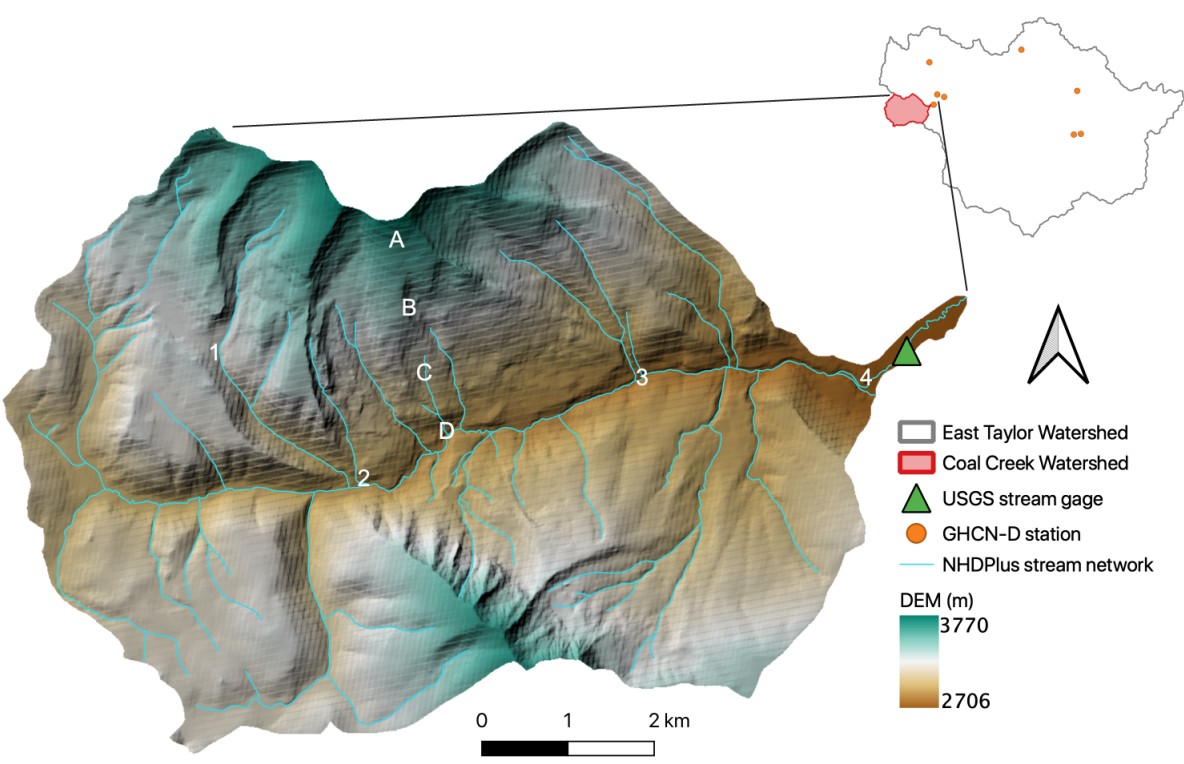

**Figure 1.** Map of Coal Creek Watershed in relation to the larger East Taylor Watershed. Also shown are the location of GHCN-D stations, USGS stream gage, NHDPlus stream network, and Digital Elevation Model (DEM). The marked points (A-D) and (1-4) are point locations for groundwater table and surface ponded depth, respectively.

The ATS meshes including surface land covers and subsurface structures and properties were developed using the Watershed Workflow package (https://github.com/ecoon/watershed-workflow), which brings together a variety of data streams, delineates the catchment, and generates a variable resolution mesh with refined resolution at the stream network. Resolutions ranged

from typical triangle areas of $5,000\ m^2$ near the stream network to $50,000\ m^2$ away from the stream network. This triangular surficial mesh was then elevated using Digital Elevation Model (DEM) from the USGS National Elevation Dataset (NED) $30\ m$ resolution dataset.

On the surface, 14 land cover types were delineated from the National Land Cover Database (NLCD 2016) product for the conterminous United States (CONUS). The Leaf Area Index (LAI) seasonal variations for each land cover type were retrieved

from MODIS (https://modis.gsfc.nasa.gov/data). Some of the plant functional type and their properties such as rooting profile and photosynthetic parameters were adopted from parameters used in the CLM 4.5 technical notes (Oleson et al., 2013).



In the subsurface, the model was discretized into 19 terrain-following layers with a total thickness of ∼28 m. A total of 6 soil layers encompassed the top 2 m of the domain. The depth to bedrock (DTB) was determined from SoilGrids (Shangguan et al., 2017) that varies from 3 m at the shallowest to 26 m at the deepest. The geologic layers were sandwiched between the soil and bedrock layers. The vertical resolution of the mesh gradually increased from 5 $cm$ at the surface to 2 $m$ at the 2 $m$ depth, and it remained constant at 2 $m$ until the bottom of the model domain at a depth of 28 $m$. The total number of cells is 171,760.

Based on the National Resources Conservation Service (NRCS) Soil Survey Geographic (SSURGO) soils database, 22 soil types were identified and mapped within the soil layer. Due to the edge-matching issues in the SSURGO soil database (Gatzke et al., 2011), the 22 soil types were regrouped into 9 types to remove the discontinuity of a soil type across soil survey area boundaries. Using a global surface geology dataset from GLobal HYdrogeology MaPS (GLHYMPS) 2.0 (Huscroft et al., 2018), 11 geologic material types were identified and mapped within the geologic layer. The spatial distribution of the soil and geological layers was shown in Figure 2. The permeability and porosity for each soil type were retrieved from the SSURGO database, and the van Genuchten parameters were determined using Rosseta v3, a pedotransfer function that relates sand, silt, and clay percentage to van Genuchten parameters (Zhang and Schaap, 2017). The permeability and porosity for each geology type were retrieved from the GLHYMPS database. Bedrock functions as a confining layer with a very small permeability of $1\times10^{-17}$ $m^2$.

The model was first run for 1,000 years with constant precipitation (∼ 850 $mm/yr$) as the cold-spinup that resulted in steady-state model outputs at the final timestep, which was then used as the initial condition for a 10-year (October 1, 2004 - October 1, 2014) transient simulation (i.e., warm-spinup) driven by the Daymet forcing. Model state at the end of the 10-year run was used as the initial condition for a 4-year transient run (October 1, 2015 - October 1, 2019) driven by various meteorological forcings. The year 2015 was treated as a second warm-spinup and was discarded from the analysis to avoid any influence from previous spinup runs. The study period features a high snow year (2017) and a low snow year (2018), allowing us to demonstrate how different meteorological forcing impact watershed responses under various weather conditions. ATS runs were taken at sub-hourly timestep determined by the model while outputting streamflow and watershed averaged variables at hourly timestep. Due to the large memory usage, spatially distributed variables such as SWE and ET were output at daily timesteps. Each run took ∼ 17 $hours$ wallclock time using 64 processors on the Cori clusters at the National Energy Research Scientific Computing Center (NERSC). The models were not calibrated because the focus of this study was to evaluate the effect of meteorological forcings on model simulation instead of estimating the optimal parameters used in ATS.

## 2.3 Gridded meteorological forcing

For this comparison, three widely used GMF were considered: PRISM (Daly et al., 2008), NLDAS-2, (Xia et al., 2012)), and Daymet v4 (Thornton et al., 1997, 2021). NLDAS-2 and Daymet v4 are hereafter referred to as NLDAS and Daymet, respectively. The detailed comparison between each meteorological dataset can be found in Table 1.

The Daymet climate forcing is a gridded, daily product with a spatial resolution of 1-km, covering continental North America, Puerto Rico, and Hawaii. It assimilates data from weather stations and accounts for elevation, prevailing winds, storm

**Figure 2.** (a) Land cover, (b) soil map, (c) geology map of Coal Creek Watershed that are generated from Watershed Workflow. (d) ATS simulated surface ponded depth and soil saturation on October 1, 2018. The zoomed in plot shows the 3D unstructured triangular mesh.



**Table 1.** Meteorological datasets comparison

| Name | Primary variables | Derived/Secondary variables | Spatial resolution | Temporal resolution | Spatial coverage | Temporal coverage | Data source | Interpolation method |
|---|---|---|---|---|---|---|---|---|
| Daymet | Tmin, Tmax, Prcp | Srad, VP, SWE, Dayl | $1 - km$ | daily | North America, Hawaii, and Puerto Rico[a] | 1980-present | meteorological stations | geographically weighted regression (Thornton et al., 1997) |
| PRISM | Tmin, Tmax, Prcp, Tdmean, Vpdmin, Vpdmax | Tmean, VP | $800\ m$[b] | daily | CONUS | 1981-present | meteorological stations | geographically and elevation-weighted regression, station weighting by topography, distance to coast, and atmospheric factors (Daly et al., 2008) |
| NLDAS | Tmean, Prcp, Srad, Lrad, SH, WS, VP | pET | $1/8th\ deg$, $\sim 12 - km$ | hourly | North America | 1979-present | National Centers for Environmental Prediction (NCEP) data assimilation fields and meteorological stations | bilinear interpolation of NCEP North America Regional Reanalysis (NARR) adjusted for elevation using PRISM methodology, temporally disaggregated to one hour (Cosgrove et al., 2003) |

Abbreviations: Tmin: minimum temperature, Tmax: maximum temperature, Tmean: mean temperature, Prcp: precipitation, Srad: shortwave radiation, Lrad: longwave radition, VP: vapor pressure, SWE: snow water equivalent, Dayl: day length, Tdmean: mean dew point temperature, Vpdmin: minimum vapor pressure deficit, Vpdmax: maximum vapor pressure deficit, SH: specific humidity, WS: wind speed, pET: potential evaporation.

[a] Puerto Rico has Daymet since 1950.

[b] this native resolution is not free.





tracks, and proximity to large water bodies (Thornton et al., 1997). Here, the latest Daymet version 4 product is used because this product has gone through significant bias corrections in station observations and the gridded product shows a better match with weather stations compared to the earlier versions (Thornton et al., 2021).

The PRISM forcing is developed by the PRISM climate group at the Oregon State University and is recognized as the

official climate dataset for the U.S. Department of Agriculture. It utilizes a wide range of monitoring networks to generate daily, spatially continuous climate data for the CONUS. The PRISM provides a native grid resolution of 30-arcsec ($\sim$800 $m$) for a fee, but also provides a coarsened $4\ km$ resolution free of charge. We used the native 30-arcsec resolution and downscaled (upscaled) the dataset to obtain finer (coarser) spatial resolutions.

The NLDAS dataset is a gridded, hourly product with a spatial resolution of 1/8th degree ($\sim$12 $km$ at the study site) for the

entire North American region. The non-precipitation forcing variables are primarily derived from the North American Regional Reanalysis (NARR) by spatially interpolating data from the 32-km resolution NARR grid to the 1/8th degree NLDAS grid while temporally disaggregated from 3-hourly to hourly frequency (Cosgrove et al., 2003). The precipitation is a product of a temporal disaggregation of a gage-only Climate Prediction Center (CPC) analysis of daily precipitation into hourly frequency, performed directly on the NLDAS grid and including an orographic adjustment based on the widely-applied PRISM climatology.

All three datasets provide temperature and precipitation as the primary forcing with a few secondary forcing variables. In addition to temperature and precipitation, ATS requires solar radiation (both incoming shortwave radiation (Srad) and longwave radiation (Lrad)), relative humidity, and wind speed as forcing inputs. Relative humidity can be estimated based on vapor pressure and mean temperature (Bolton, 1980). Lrad can be estimated from Srad and relative humidity. Because PRISM does not provide Srad and Lrad, we used solar radiation from Daymet instead. Wind speed was assumed to be constant (i.e.,

$4\ m/s$) for both Daymet and PRISM. Compared to PRISM and Daymet, NLDAS provides the most complete set of variables to drive ATS simulations.

Different meteorological forcings have different definitions for a calendar day and they are often different from the local time used in the observation data. Time zone adjustment and lag corrections have been applied to account for the time lag difference between meteorological forcing and local gages. For example, PRISM lags Daymet by one day, so PRISM has

been shifted forward one day to be consistent with Daymet. Both model simulation and gage observation have been converted to Coordinated Universal Time (UTC) timezone for hourly streamflow comparison. Please refer to Table A1 for detailed comparison of calendar day definitions. For consistency, all simulated streamflow are in hourly resolution and are compared to hourly USGS streamflow in the results section 3.

To study the effect of spatial resolution of meteorological forcing, precipitation, and temperature from 800m PRISM and

1 km Daymet have been downscaled (upscaled) into finer (coarser) spatial resolutions. The downscaling of 800 $m$ PRISM or 1 km Daymet into 400 $m$ used a data-driven downscaling approach. Specifically, Random Forests (Breiman, 2001) were used to extract the relationships between precipitation (or average temperature) and topography. These relationships were developed at 800 $m$ (for PRISM) and 1km (for Daymet) resolutions and were used as-is to generate the 400 $m$ downscaled estimates. The downscaled precipitation grids were additionally filtered to ensure a smooth field in low-gradient areas without

affecting high-gradient areas (Daly et al., 2008). The topographic variables considered were elevation, slope, aspect, latitude,





and longitude. These variables were extracted from the NED 10 m resolution product and upscaled to $400\ m$ and $800\ m$ (for PRISM) via bilinear interpolation. Upscaling of topographic variables was done in maximum increments of 2x (e.g., $10\ m->20\ m->40\ m$ and so on).

For consistency, spatial upscaling of 800 m PRISM into 1600 m and 4000 m was performed using a coarsen function from python package–`xarray` (http://xarray.pydata.org) by applying moving average based on a 2x2 window size. The same approach was used for spatial upscaling of 1 km Daymet to 2 km and 4 km. To study the effect of temporal resolution of meteorological forcing, the hourly NLDAS dataset was aggregated into 12-hourly and daily time steps by using `xarray`'s resampling method by taking the mean (for temperature, relative humidity, and solar radiation) or sum (for precipitation) for the aggregated period.

In ATS, meteorological forcing is distributed linearly across its temporal resolution, and each model surface cell gets its meteorological forcing through spatially bilinear interpolation. For example, both Daymet and PRISM apply their meteorological forcing at the daily time scale, whereas NLDAS applies its meteorological forcing at an hourly time scale. As a result, Daymet and PRISM show less temporal dynamics compared to NLDAS.

## 2.4  Observation data

Instantaneous streamflow data (every 15 minutes) are available from April 1st through November 15th every year since 2014 through a USGS gage (station number: 09111250), located at the watershed outlet. Past Airborne Snow Observatory (ASO) survey has four flights covering this watershed in 2018 and 2019 to survey the snow depth and SWE. Remote sensing products such as the Moderate Resolution Imaging Spectroradiometer (MODIS) 8-day composite ET is available at a 500 m resolution since 2000.

To compare the accuracy of each meteorological forcing against field observations, all three meteorological forcings in their native resolutions were compared against GHCN-D weather stations within the East Taylor Watershed. In total, there were seven stations with long-term precipitation records and four stations with long-term temperature records (see GHCN-D station locations in Figure 1). Both precipitation and temperature time series were extracted at each GHCN-D gage location from the GMF.

## 2.5  Model evaluation metrics

Model simulated outputs were compared against observation data including hourly streamflow from a USGS gage and spatially distributed SWE from the ASO survey. The modified Kling-Gupta efficiency ($KGE$) and its three components ($r, \gamma, \beta$) were used to evaluate the model performance (Kling et al., 2012).

$$KGE = 1 - \sqrt{(r-1)^2 + (\gamma-1)^2 + (\beta-1)^2} \tag{5}$$





with:

$$r = \frac{cov(S,O)}{\sigma_s \sigma_o} \tag{6}$$

$$\gamma = \frac{\sigma_s / \mu_s}{\sigma_o / \mu_o} \tag{7}$$

$$\beta = \frac{\mu_s}{\mu_o} \tag{8}$$

where $S$ and $O$ represents simulated and observed values, respectively, $r$ is correlation coefficient, $\gamma$ is variability ratio, $\beta$ is
the bias ratio, $cov(S,O)$ is the covariance between simulated and observed values, $\sigma$ is the standard deviation, $\mu$ is the mean.

Using the modified $KGE$ avoids the effect of input bias on the variability indicator which has an advantage over the original
$KGE$ (Gupta et al., 2009; Kling et al., 2012), and it also allows diagnostic interpretation of the performance score. $KGE$
decomposes model performance into correlation ($r$), variability ($\gamma$) and bias ($\beta$) term. For example, the correlation measures
the temporal dynamics of streamflow while the variability and bias measure the flow duration curve. The $KGE$ ranges from
$-\infty$ (poorest model skill) to 1 (perfect) when all three terms reach unity.

Taylor Diagram is used to show how close a set of patterns (e.g., meteorological forcing) match observations (Taylor, 2001).
On each Taylor Diagram, performance metrics such as standard deviation and Pearson's correlation coefficient ($r$) are shown
together. The azimuthal angle represents correlation, and the radial distance represents the standard deviation. Also can be
shown is the centered Root Mean Square Error ($RMSE$) between simulation and observation. The relationship between these
statistics was shown below:

$$E^2 = \sigma_s{}^2 + \sigma_o{}^2 - 2\sigma_s \sigma_o r \tag{9}$$

where $E$ is the centered $RMSE$, which is also measured by the geometric distance between simulation and observation
data points on the Taylor Diagram (unit is the same as standard deviation). In cases where more than one observation point are
plotted on the same diagram, the centered $RMSE$ is omitted. Note that the centered $RMSE$ is a mean-removed $RMSE$, and
thus any bias in the data is not shown.

The closer the distance between simulation and observation data point on a Taylor Diagram, the smaller the centered $RMSE$
(observation data point has centered $RMSE = 0$), the more similarity they show in terms of standard deviation, and the higher
the correlation coefficient (observation data point has $r = 1$).





## 3   Results

### 255   3.1   Comparison between meteorological forcing and weather stations

Taylor Diagram was used to compare the similarity in precipitation and temperature patterns between meteorological forcing and GHCN-D stations (Figure 3). Compared to temperature, precipitation showed stronger spatial heterogeneity among stations indicated by the larger difference in standard deviation and correlation. The close clustering of temperature data points indicated that the difference between different stations in temperature patterns was small. For precipitation, PRISM showed a strong

correlation ($r > 0.9$) with GHCN-D at three stations, whereas Daymet only showed a strong correlation at one location and all NLDAS sites showed a relatively weak correlation ($0.6 < r < 0.8$). For temperature, all three meteorological forcings showed a very strong correlation ($r > 0.95$) with GHCN-D, though PRISM and Daymet were slightly better than NLDAS. It is not surprising that Daymet and PRISM have very close resemblance since they used similar weather stations as their main forcing input (Thornton et al., 1997; Daly et al., 2008). Previous studies also reported similar findings at different watersheds that

Daymet and PRISM showed better agreement with ground-based observational data than NLDAS (Muche et al., 2020), and the temperature was more accurately represented than precipitation (Behnke et al., 2016).

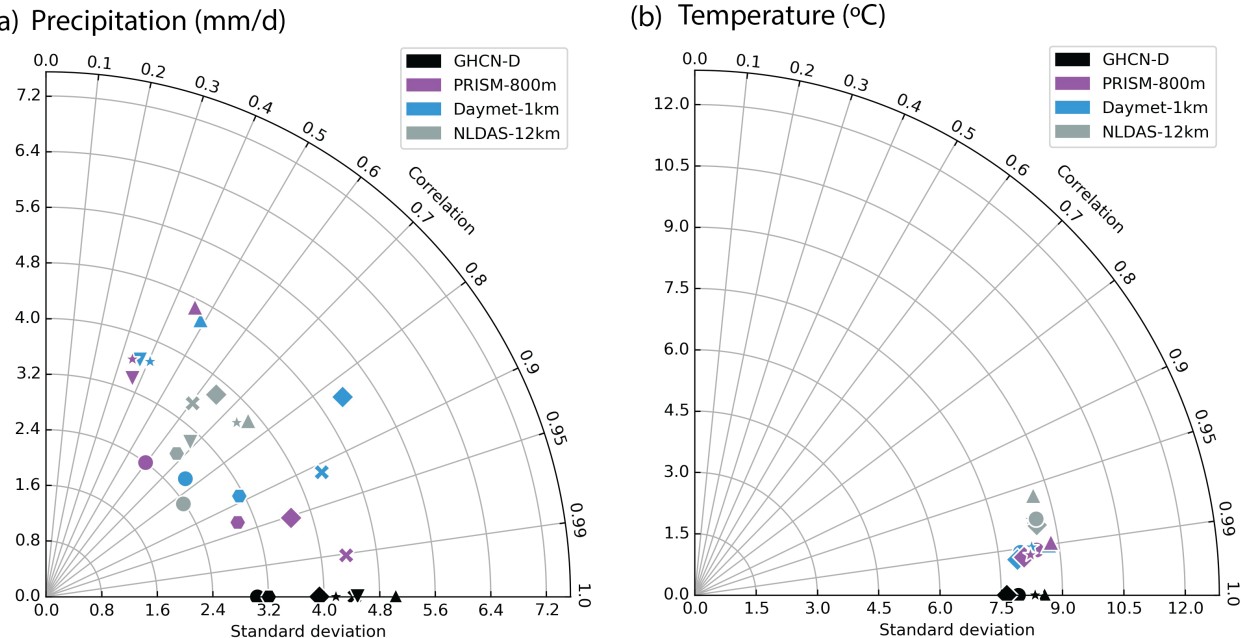

**Figure 3.** Taylor Diagram showing the correlation coefficients and standard deviations between meteorological forcing and GHCN-D gages (black) for (a) precipitation and (b) temperature. The azimuthal angle represents correlation, and the radial distance represents the standard deviation. Each marker symbol represents a different GHCN-D station location.





## 3.2 ATS simulations driven by different meteorological forcing products

Simulated hourly streamflow forced by Daymet (1 km, daily) showed better performance against USGS hourly streamflow, followed by PRISM (800 m, daily) and NLDAS (12 km, hourly) during a three year simulation period (Figure 4). In their
native resolutions, Daymet outperformed PRIMS and NLDAS with the largest $KGE$ (0.62) against observed streamflow (also see the statistical summary in Table 2). The high agreement between observed and simulated streamflow forced by Daymet is remarkable given that the ATS model has not been calibrated. In general, all three models underestimated discharge ($\beta < 1$) while showing slightly more variability ($\gamma > 1$), especially in 2018 and 2019. As expected, models using hourly NLDAS showed larger variability that preserved the diurnal cycle in simulated streamflow compared to those using daily Daymet or
PRISM (Figure 5). For example, Daymet underestimated peak flows but it showed larger flow variability during low flows and early spring. In addition, NLDAS showed a large time lag ( 3 days in 2018 and  8 days in 2019) in peak flows. during early spring and summer.

**Table 2.** Summary statistical metrics for evaluation of streamflow at watershed outlet under different meteorological forcing compared to observed streamflow from USGS

| Meteorological forcing | Spatial resolution | Temporal resolution | $r$ | $\gamma$ | $\beta$ | $KGE$ |
|---|---|---|---|---|---|---|
| PRISM | 400 m | daily | 0.70 | 1.01 | 0.53 | 0.44 |
| | 800 m | daily | 0.70 | 1.02 | 0.54 | 0.45 |
| | 1600 m | daily | 0.68 | 1.08 | 0.55 | 0.44 |
| | 4000 m | daily | 0.64 | 1.20 | 0.56 | 0.40 |
| Daymet | 400 m | daily | 0.76 | 1.04 | 0.71 | 0.63 |
| | 1 km | daily | 0.75 | 1.05 | 0.72 | 0.62 |
| | 2 km | daily | 0.73 | 1.10 | 0.72 | 0.60 |
| | 4 km | daily | 0.73 | 1.14 | 0.73 | 0.59 |
| NLDAS | 12 km | hourly | 0.60 | 1.74 | 0.27 | -0.11 |
| | 12 km | 12-hourly | 0.69 | 1.59 | 0.35 | 0.07 |
| | 12 km | daily | 0.73 | 1.47 | 0.43 | 0.22 |





**Table 3.** Summary of statistical metrics used for evaluation of spatially distributed SWE under different meteorological forcing at different times compared to observed SWE from ASO

| Date | Meteorological forcing | Spatial resolution | Temporal resolution | $r$ | $\gamma$ | $\beta$ | $KGE$ |
|---|---|---|---|---|---|---|---|
| March 31, 2018 | PRISM | 400 m | daily | 0.51 | 1.04 | 0.00 | -0.12 |
| | PRISM | 800 m | daily | 0.50 | 1.04 | 0.00 | -0.12 |
| | PRISM | 1600 m | daily | 0.46 | 0.94 | 0.00 | -0.13 |
| | PRISM | 4000 m | daily | 0.37 | 0.86 | 0.00 | -0.18 |
| | Daymet | 1 km | daily | 0.47 | 1.11 | 0.00 | -0.14 |
| | NLDAS | 12 km | hourly | 0.15 | 1.00 | 0.00 | -0.31 |
| May 24, 2018 | PRISM | 400 m | daily | 0.55 | 1.53 | 0.00 | -0.22 |
| | PRISM | 800 m | daily | 0.51 | 1.54 | 0.00 | -0.24 |
| | PRISM | 1600 m | daily | 0.47 | 1.77 | 0.00 | -0.37 |
| | PRISM | 4000 m | daily | 0.28 | 2.92 | 0.00 | -1.28 |
| | Daymet | 1 km | daily | 0.35 | 3.41 | 0.00 | -1.69 |
| | NLDAS | 12 km | hourly | -0.01 | 1.14 | 0.00 | -0.43 |
| April 7, 2019 | PRISM | 400 m | daily | 0.55 | 0.62 | 0.00 | -0.16 |
| | PRISM | 800 m | daily | 0.53 | 0.61 | 0.00 | -0.17 |
| | PRISM | 1600 m | daily | 0.50 | 0.57 | 0.00 | -0.20 |
| | PRISM | 4000 m | daily | 0.37 | 0.48 | 0.00 | -0.29 |
| | Daymet | 1 km | daily | 0.47 | 0.58 | 0.00 | -0.21 |
| | NLDAS | 12 km | hourly | -0.04 | 0.49 | 0.00 | -0.53 |
| June 10, 2019 | PRISM | 400 m | daily | 0.76 | 1.62 | 0.00 | -0.20 |
| | PRISM | 800 m | daily | 0.71 | 1.61 | 0.00 | -0.21 |
| | PRISM | 1600 m | daily | 0.66 | 1.48 | 0.00 | -0.16 |
| | PRISM | 4000 m | daily | 0.50 | 0.82 | 0.00 | -0.13 |
| | Daymet | 1 km | daily | 0.73 | 1.23 | 0.00 | -0.06 |
| | NLDAS | 12 km | hourly | -0.13 | 1.75 | 0.00 | -0.69 |

Average SWE across the watershed showed large differences between different meteorological forcings (Figure 6). Daymet had the largest simulated SWE on average, while NLDAS had the smallest simulated SWE. PRISM produced a similar SWE pattern with Daymet, and their magnitude was very close except for the year 2017. The accumulation of SWE started around the same time for all three meteorological forcing, though SWE disappeared early for NLDAS in the last two years.

Spatially, all meteorological forcing significantly underestimated SWE ($\beta \approx 0$) when compared to SWE from ASO (Table 3), and the difference becomes larger in higher elevation with more accumulated snow (Figure 7). The large difference may





be attributed to the higher spatial resolution (50 m) used in the ASO snow survey than the spatial resolution of the GMF.
Simulated SWE by the three models showed a larger variability ($\gamma > 1$) than the observed SWE, except on April 7, 2019 when variability was smaller than the observed SWE. Interestingly, the largest variability occurred on May 24, 2018 when little snow was accumulated on the surface. Most of the time, PRISM showed a higher correlation between simulated and observed SWE than that from Daymet and NLDAS. In contrast, NLDAS showed the poorest correlation ($r < 0.2$) with observation.

NLDAS hourly forcing resulted in more dynamic fluctuations in the simulated watershed average ET than those from daily
Daymet and PRISM forcing (Figure 6 and Figure 5). Daymet and PRISM have almost identical ET because PRISM used the same shortwave radiation from Daymet. Compared to the remote sensed 8-day composite ET from MODIS, all three meteorological forcing showed a consistent seasonal trend with the MODIS with underestimated ET in the spring. Additionally, the simulated 8-day composite ET by Daymet and PRISM was higher than that from NLDAS in the peak growing season in 2017 and 2019.

## 3.3 Effects of meteorological forcing spatial resolution

To evaluate the effects of spatial resolution of meteorological forcing, precipitation, and temperature, different spatial resolutions of PRISM and Daymet were used to drive the model. Because the findings from both PRISM and Daymet were similar, only results from PRISM were summarized below (see Supplemental Information for Daymet comparison Figure A1 and A2). As the spatial resolution became finer, the spatial pattern of precipitation and temperature became more heterogeneous and were
more strongly associated with local topography, land use, and land cover. On the contrary, coarser-resolution meteorological forcing produced more homogeneous and smoother spatial patterns with less accuracy (Figure A3 and A4).

The simulated discharge showed similar performance in terms of KGE compared to the observation when meteorological forcing spatial resolution was $<= 1600\ m$ (Table 2). The KGE between 400m resolution and 800m resolution were almost identical. All four resolution showed higher variability ($\gamma > 1$) and high correlation ($r > 0.6$) in simulated streamflow than the
observation. The variability became larger ($\gamma = 1.20$) and the correlation ($r = 0.64$) was weaker when meteorological forcing spatial resolution reached 4 km (Figure 8). The simulated SWE and total water storage changes were almost identical for all spatial resolutions except during the snowmelt period when the 4 km spatial resolution shows faster snowmelt in early summer across all three years. The spatial distribution of SWE when compared with ASO SWE showed significantly large bias ($\beta \approx 0$) and thus negative KGE for all spatial resolution at all time (Figure 10 and Table 3). Generally, the 4 km resolution had the
worst performance and became most obvious on May 24, 2018. PRISM at 400 m and 800 m showed a similar spatial pattern thus a similar correlation with ASO SWE.

Soil moisture at the top 5cm layer showed similar pattern when spatial resolution was $<= 1600m$ (Figure A5). The differences between 4 km resolution and finer resolution became obvious during the snowmelt period (May 24, 2018 and June 10, 2019) when soil becomes saturated. For example, soil in the northwest region from the 4 km resolution was wetter on May 24,
2018, whereas soil close to the outlet from the 4 km resolution was wetter on June 10, 2019. Similarly, spatially distributed ET did not show a significant difference until meteorological forcing resolution reached 4 km (Figure A6).



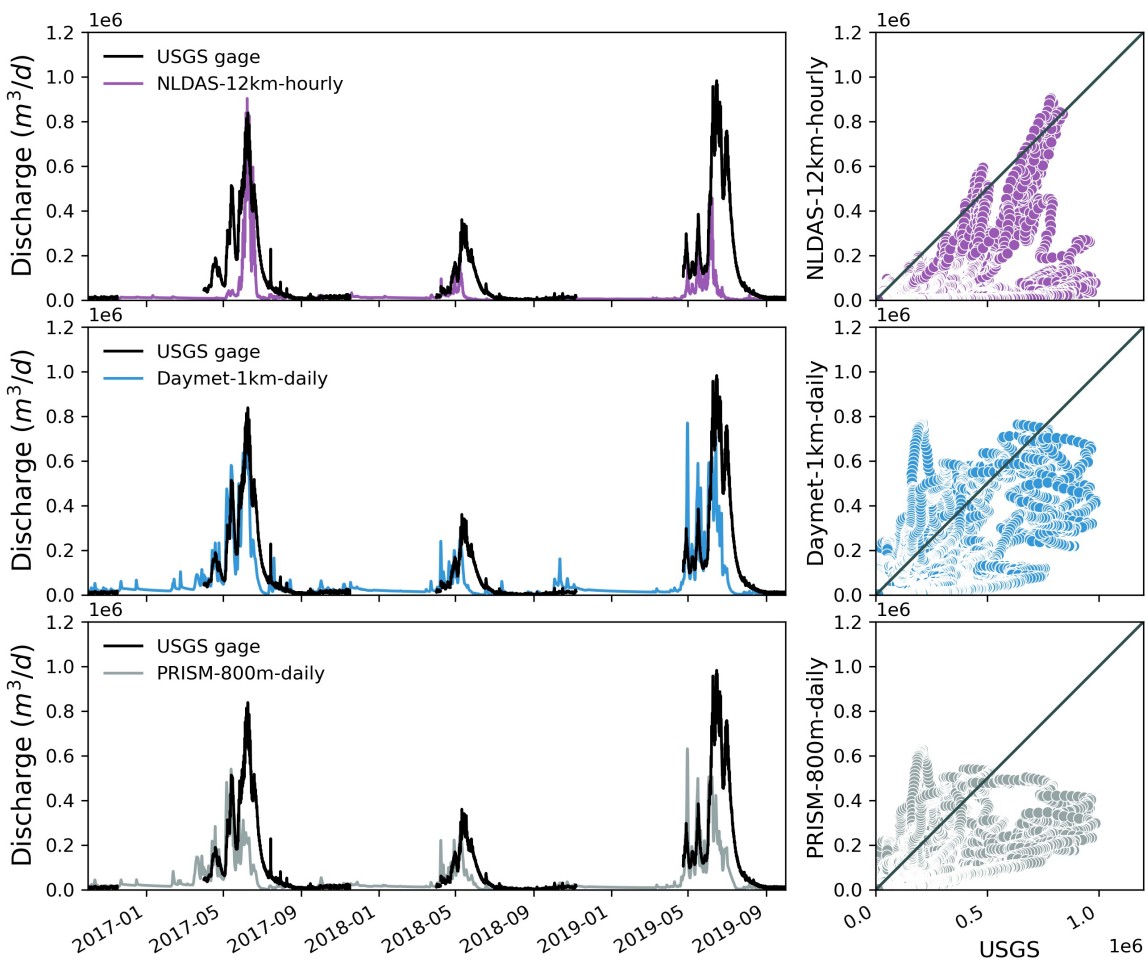

**Figure 4.** Simulated hourly discharge at watershed outlet compared with USGS hourly streamflow under different meteorological forcing. Also shown is the one to one plot..





**Figure 5.** Zoomed in plot showing simulated hourly watershed average SWE, ET and discharge under different meteorological forcing from June 1st, 2017 to June 29th, 2017.

**Figure 6.** Simulated hourly watershed average SWE and ET under different meteorological forcing. Also shown is the comparison between MODIS and simulated 8-day composite ET.


**Figure 7.** Spatial distribution of SWE under different meteorological forcing, and their comparison with ASO SWE data at four different survey times.

Surface ponded depth showed very little difference between different spatial resolutions. Four locations (labeled as 1-4 in Figure 1) were selected across the watershed to show the ponded depth variations (Figure 11). One was located at the upstream branch and the other three were located along the Coal Creek main stem. Similar to the watershed discharge, ponded depth only differed when the spatial resolution was coarsened to 4 km. The 4 km resolution had a faster recession during peak flows compared to results from other finer resolutions, and the 4 km resolution was also less responsive to rainfall. On average, surface ponded depth varied less than 0.2 m during peak flows.

A transect was selected running from mountain top to river valley bottom with four selected observation locations (labeled as A-D in Figure 1) and groundwater table time series was plotted in Figure 12. In general, groundwater rose during snowmelt (April to June) and rainfall events and fell in the dry period (July to September). Location A was at the mountain top and the groundwater table was mostly 4 m below the land surface except during the snowmelt period when the groundwater table



rose to the surface. Since location A was dominated by snow and receives less rainfall, the rise in the groundwater table was mainly due to snowmelt. Location B and C showed similar trends in groundwater table fluctuations, however, they showed more peaks since they were influenced by both snowmelt and rainfall. The groundwater table at location D was mostly close to the surface except during the dry season when groundwater started to decline. The 4 km resolution behaved very differently from the other finer resolutions at location A, where the groundwater table peaked earlier due to earlier infiltration from snow. In a dry year in 2018, the groundwater table did not even rise during the snowmelt period at location A. In general, the coarser the meteorological forcing resolution, the larger the bias in precipitation, and the more the groundwater table buffered from snowmelt and rainfall.

### 3.4 Effects of meteorological forcing temporal resolution

The effect of meteorological forcing temporal resolution on watershed hydrologic simulations was evaluated by using hourly, 12-hourly, and daily NLDAS datasets. All resolutions outputted hydrological variables in hourly timestep and were compared with hourly streamflow.

The match between simulated and observed hourly discharge in terms of $KGE$ was better with daily resolution than hourly and 12-hourly temporal resolution (see Figure 13 and Table 2). Across different temporal resolutions, the correlation between simulated and observed streamflow was relatively high ($r > 0.6$), however, the simulated flow was underestimated ($\beta < 0.5$) and showed more variability ($\gamma > 1.4$) than the observed flow. The hourly NLDAS had the highest correlation ($r = 0.73$) and lowest bias ($\beta$). This is not surprising since hourly meteorological forcing has a more dynamic forcing pattern including hourly temperature and precipitation, and thus it yielded a more dynamic overland flow pattern that can be quite different from field observations. To investigate the hydrograph in more detail, discharge time series were zoomed into the high flow season in 2017. It was clear that models driven by hourly and 12-hourly NLDAS contained sub-daily flow fluctuations that would be absent in models driven by daily NLDAS (Figure 14). Models driven by 12-hourly NLDAS had a weaker diurnal flow pattern in terms of magnitude, and its peaks were asynchronous from the USGS streamflow. This is because hourly NLDAS retains the diurnal signal of air temperature and solar radiation and thus the diurnal snowmelt pattern, whereas 12-hourly NLDAS had a weaker diurnal signal due to only changing air temperature and solar radiation every 12 hours (Figure 14). On the other hand, daily NLDAS assumes uniform air temperature and solar radiation throughout a day, and thus produced streamflow that was much smoother without diurnal fluctuations. The aggregation of temporal resolution of meteorological forcing led to a time shift in flow peaks by several hours. In general, discharge from the hourly NLDAS peaked earlier and had less steep recession limb compared to that from 12-hourly and daily NLDAS.

Hourly NLDAS had the largest snowmelt and ET variations, followed by 12-hourly and daily NLDAS (Figure 14). The large ET from hourly NLDAS was due to the strong solar radiation at an hourly rate. Compared to the remote sensed 8-day composite ET from MODIS, the simulated 8-day composite ET forced by the different temporal resolution of NLDAS showed a consistent seasonal trend with the MODIS (Figure 15). However, significant model underestimation was observed in the spring. All three NLDAS resolutions showed similar trends of seasonal snowpack accumulation (Figure 15). However, there

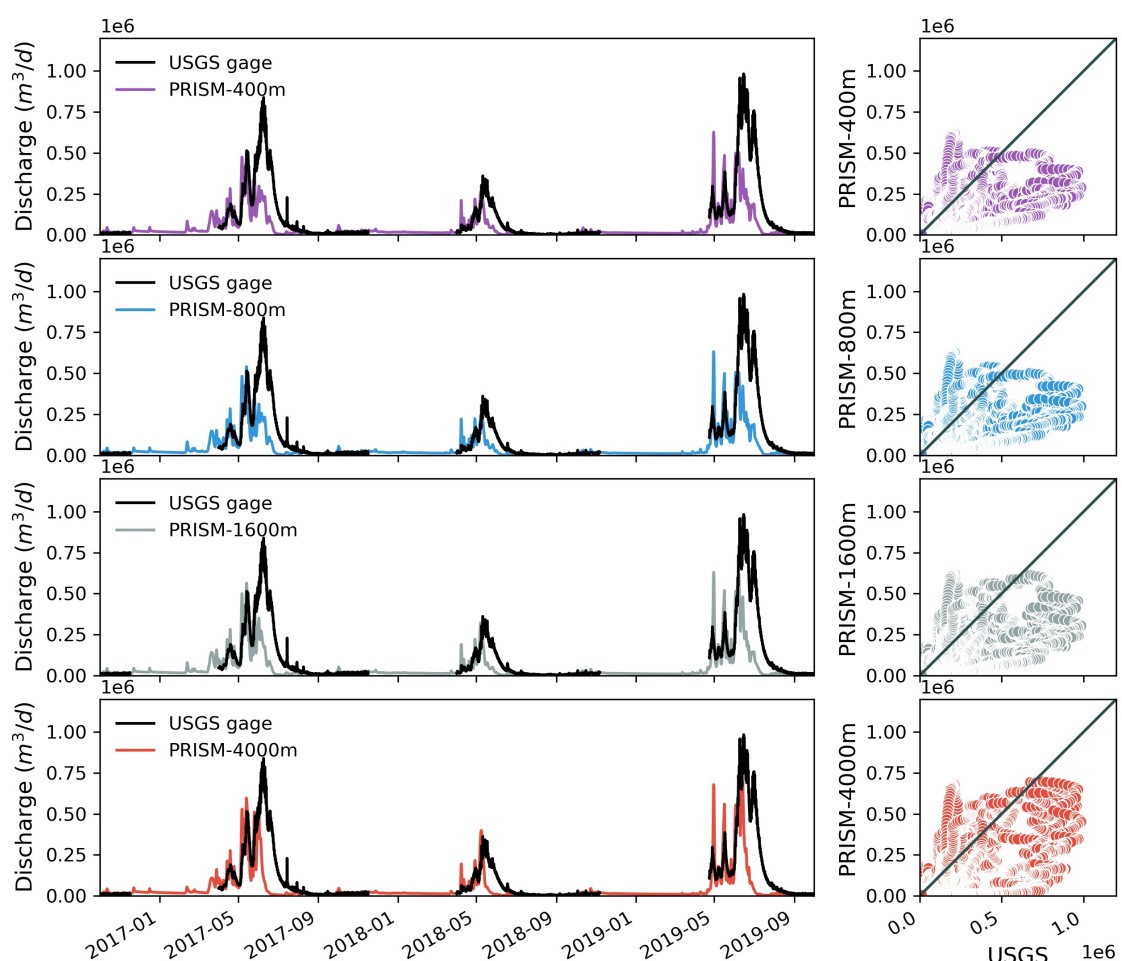

**Figure 8.** Simulated discharge at watershed outlet compared with USGS gage under different spatial resolution of PRISM. Also shown is the one to one plot.

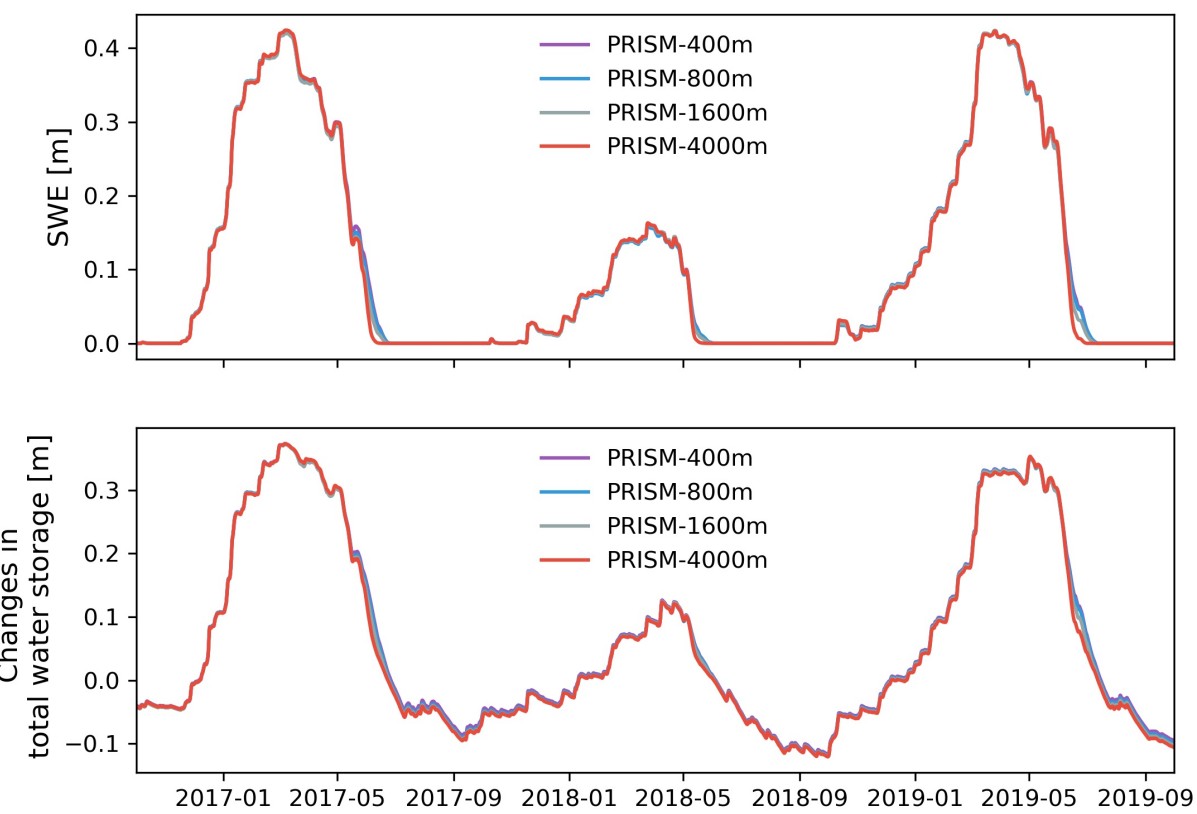

**Figure 9.** Simulated watershed average SWE and changes in total water storage under different spatial resolution of PRISM.

was a large difference in peak SWE. Hourly NLDAS reached a smaller SWE peak and the snow melted earlier compared to

12-hourly and daily NLDAS, which might be the combined effects of faster snowmelt and larger ET. Additionally

## 4   Discussions

### 4.1   The choice of gridded meteorological forcing for integrated watershed simulation

We compared three GMFs, Daymet-1 km, PRISM-800 m and NLDAS-hourly, in terms of watershed outlet discharge, spatially

distributed variables including SWE and ET. Can we choose the "best" meteorological forcing for integrated watershed mod-

eling based on discharge comparison alone? What are the strengths and weaknesses of each meteorological forcing? In surface





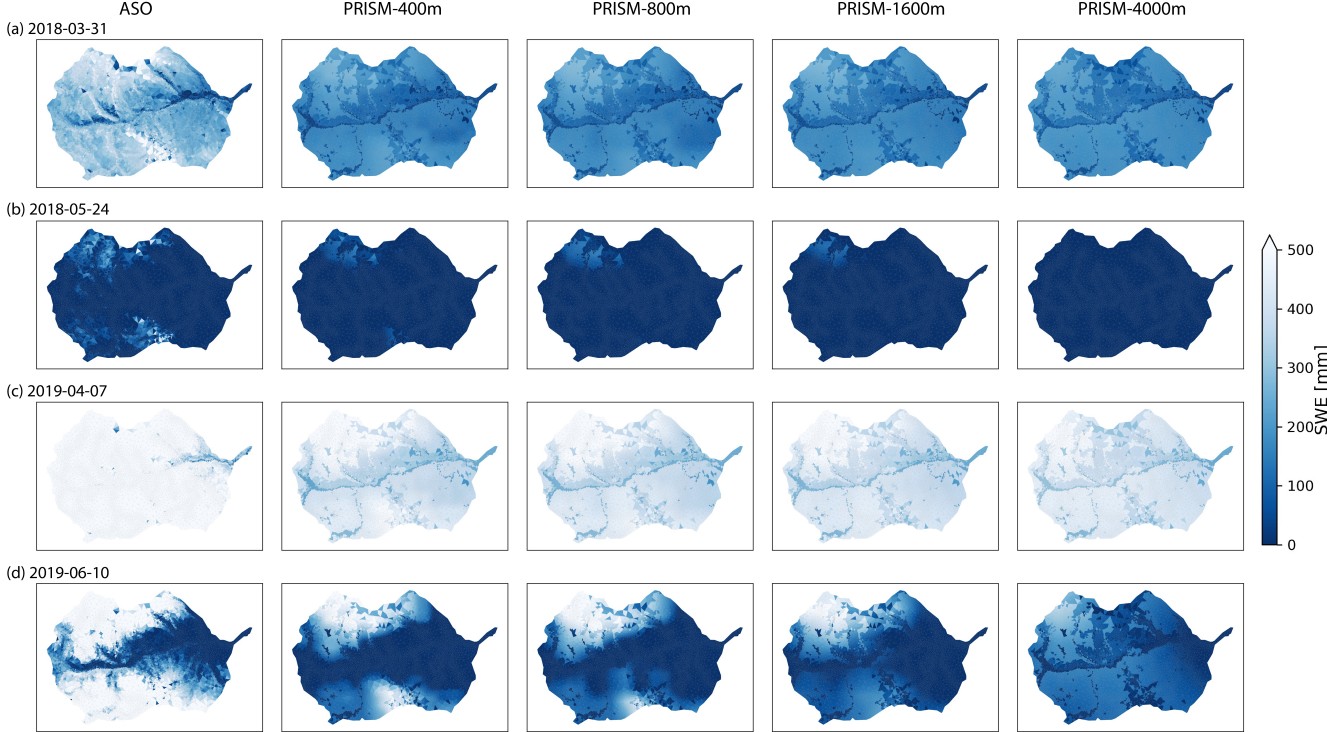

**Figure 10.** Spatial distribution of SWE under different spatial resolution of PRISM, and their comparison with ASO SWE data at four different survey times.

hydrologic modeling, hydrologists often judge the performance of a numerical model by the ability to match the streamflow at the watershed outlet, and (Staudinger et al., 2019). However, streamflow alone is not good enough for evaluating the performance in meteorological forcing because discharge at the watershed outlet has limited information on the spatial distribution

of model outputs (e.g., SWE). Even though simulated streamflow from Daymet has the best match (i.e., highest $KGE$) against observation, the spatially distributed SWE from Daymet has a weaker correlation with the observed SWE from ASO than that from PRISM. Both Daymet and PRISM perform better than NLDAS in simulating discharge and spatial SWE due to their relatively fine spatial resolution. As shown in Figure 7, NLDAS hardly captures the spatial heterogeneity of SWE when comparing to ASO SWE. The entire watershed area ($\sim53.2\ km^2$) is smaller than the size of one pixel of the NLDAS grid ($\sim12 \times 12\ km$

or $\sim 144\ km^2$), making the meteorological forcing almost homogeneous at the watershed scale.

For a watershed simulation that usually has a mesh resolution $< 1km$, Daymet or PRISM provides the best spatial resolution available across the U.S. However, they do not have the complete forcing dataset that could be directly applied to watershed models without filling in the missing data set. Daymet has most of the forcing variables except wind speed and longwave radiation. PRISM is missing both wind speed and solar radiation (i.e., Srad and Lrad), which is important for calculating

surface energy balance and estimating ET. A common approach is to fill in the missing variables using a different source. For

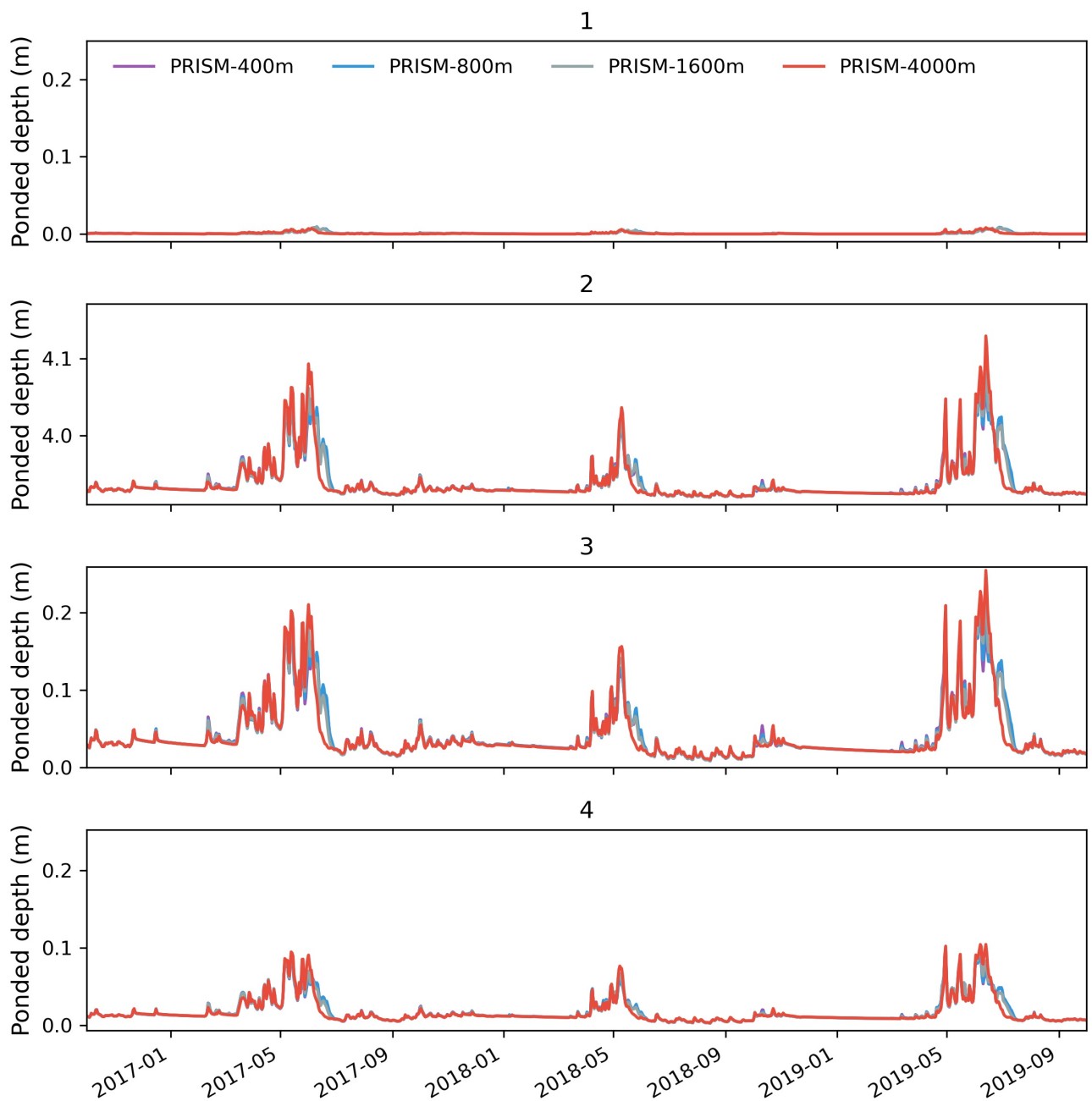

**Figure 11.** Simulated surface ponded depth at four selected locations under different spatial resolution of PRISM.

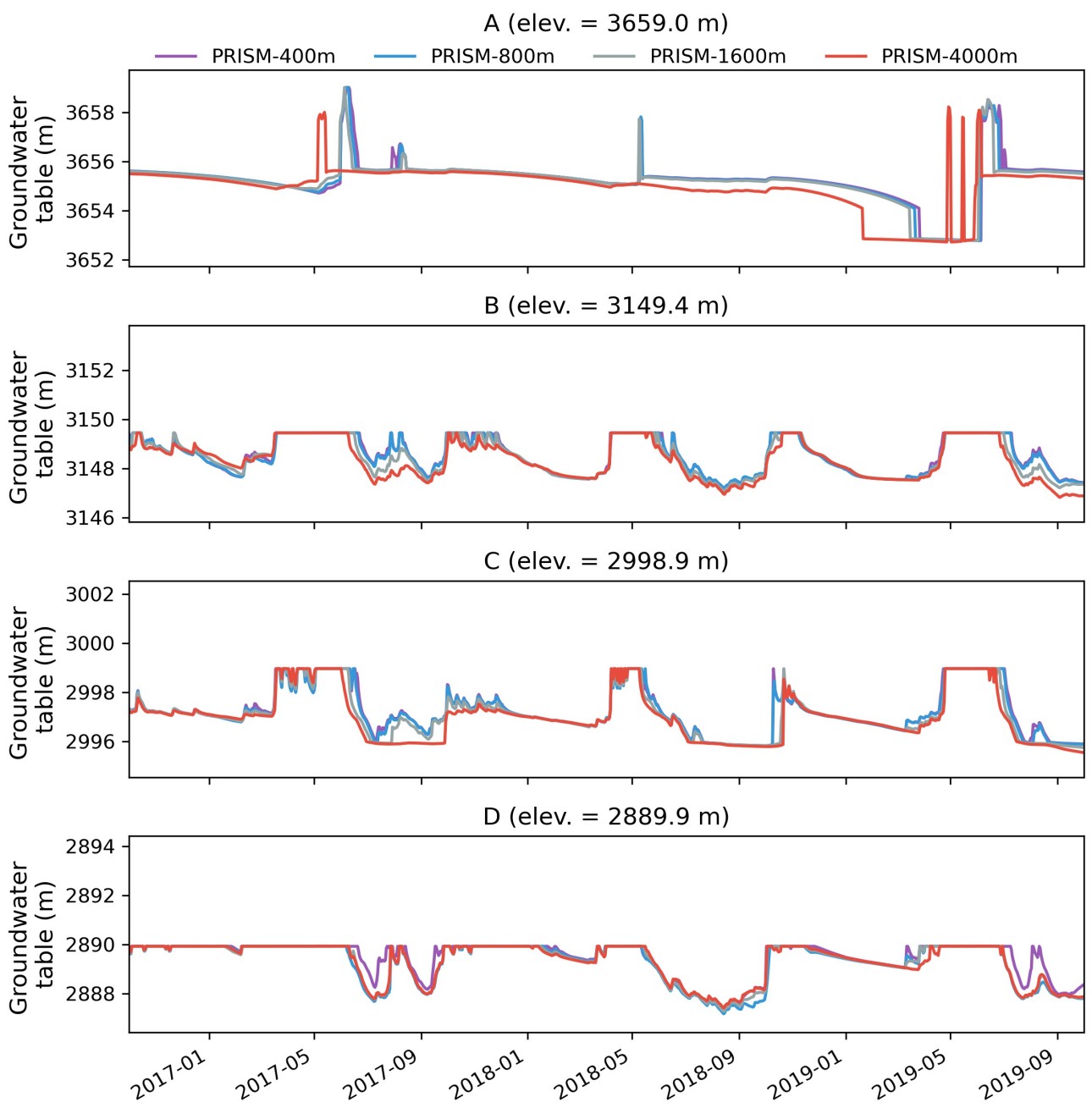

**Figure 12.** Simulated groundwater table at four selected locations (A-D) under different spatial resolution of PRISM. Also see Figure 1 for location detail.

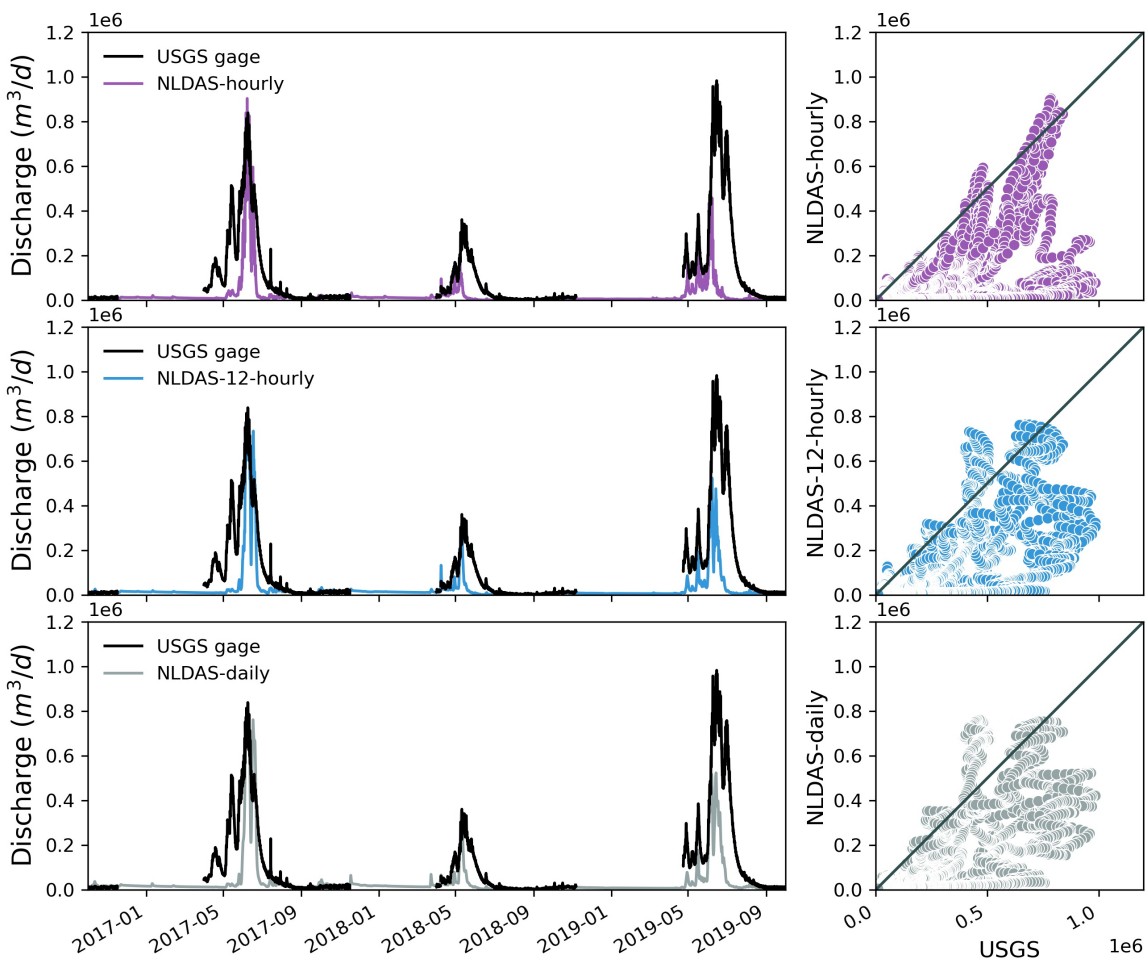

**Figure 13.** Simulated hourly discharge at watershed outlet compared with hourly USGS streamflow under different temporal resolution of NLDAS. Also shown is the one to one plot with metrics.





**Figure 14.** Zoomed in plots showing hourly simulated snowmelt, ET, and discharge under different temporal resolution of NLDAS in a high flow season in 2017.

**Figure 15.** Simulated watershed average SWE and ET under different temporal resolution of NLDAS. Also shown is the comparison between MODIS and simulated 8-day composite ET.





example, Mourtzinis et al. (2017) used solar radiation from the National Aeronautics and Space Administration's POWER (NASA-POWER) database ($12,000 \; km^2$ resolution) combined with PRISM's temperature and precipitation to simulate a crop model. In our study, we used Daymet as a source of solar radiation for PRISM, and the model was able to simulate streamflow reasonably well ($KGE = 0.45$) (Table 2) while capturing the spatial heterogeneity of distributed variables.

On the other hand, NLDAS provides the most complete forcing dataset with hourly resolution but comes with a spatial resolution of $\sim 12 \; km$ which is too coarse for watershed simulation, especially watershed with complex terrain. The spatial resolution may become less important as model resolution becomes coarser and the focus is on system-scale water budget. For example, NLDAS at its native resolution has been applied successfully at the continental scale to study transpiration partitioning using ParFlow-CLM at 1 km resolution (Maxwell and Condon, 2016). Past studies have also attempted downscaling of NLDAS

to much finer spatial resolution (Ko et al., 2019; Pan et al., 2016). For example, Ko et al. (2019) downscaled the meteorological forcing variables from the 12-km resolution of NLDAS to 1-km resolution using high-resolution terrain information at the Río Sonora Basin. There has been other attempt to merge the high-spatial resolution of PRISM dataset with the NLDAS dataset which produced a complete forcing dataset of daily high ( 4 km) resolution covering the CONUS (Abatzoglou, 2013).

Ideally, we want to choose the GMF with the finest spatiotemporal resolution while providing the most complete forcing.

None of the three GMFs are perfect. An alternative approach is to use meteorological forcing outputted from climate models such as WRF. It has the flexibility of generating much finer spatial and temporal resolution output while providing all available meteorological forcing. Maina et al. (2020) used the nested-domain configuration of WRF to dynamically generate meteorological forcing variables at various spatial resolutions (from 0.5 km to 13.5 km) for use with ParFlow-CLM. Although forcing generated by WRF provides a viable option for meteorological input in the hydrologic model, it requires more effort to simulate

the forcing using WRF than directly using the publicly available gridded forcing (e.g., Daymet).

## 4.2    Spatial vs temporal resolution: which one is more important?

Is there an optimal spatiotemporal resolution of meteorological forcing for driving watershed simulation while producing realistic results? Should we choose finer spatial resolution over finer temporal resolution? Depending on the quantity of interest and the spatial and temporal scale of the study, the choice may differ. In this study, watershed outlet discharge is shown to be

less sensitive to both the spatial and temporal resolution of meteorological forcing because it is an accumulative quantity. The simulated discharge is almost identical between PRISM 400 m, 800 m, and 1600 m resolution. The simulated discharge only becomes noticeably worse when the spatial resolution of meteorological forcing is coarsened to 4000 m (or 30% of the watershed area). Similarly, the watershed average SWE, ET, total water storage do not show a significant difference between different spatial resolutions of PRISM and Daymet.

The spatial resolution becomes more important if the quantity of interests are the spatial distributed hydrologic variables. For example, the SWE distribution under 400 m and 800 m PRISM resembles more closely with ASO SWE than results under much coarser spatial resolution. Maina et al. (2020) also found the SWE distribution to be sensitive to the spatial resolution of meteorological forcing, with the finer resolution being able to accurately reproduce SWE spatial distribution as well as total SWE volume. In addition, a higher spatial resolution of forcing would preserve the spatial heterogeneity of distributed





variables better and may provide better estimates of variables at point locations (e.g., SWE at the NRCS's Snowpack Telemetry (SNOTEL) stations and groundwater table at wells). We show that groundwater table at high elevations can be quite different under different spatial resolutions of meteorological forcing (Figure 12).

       Temporal resolution becomes more important than spatial resolution if simulating storm events or flash floods that happen within several hours, resulting in a sharp increase of stream discharge Ochoa-Rodriguez et al. (2015). The regular, periodic

streamflow fluctuations induced by sub-daily snowmelt or ET could also impact the hyporheic exchange between surface water and groundwater (Loheide and Lundquist, 2009), which in turn impact nutrient cycling in the stream and hyporheic zone biogeochemical processes (Shuai et al., 2017; Song et al., 2018). On the contrary, it is challenging to match the simulated variable from high temporal resolution with field observation. We show that the performance in simulated discharge deteriorates when temporal resolution increases from daily to hourly using NLDAS (Figure 13). Additionally, hourly meteorological forcing

is difficult to obtain and may be subject to large bias and errors. There are also sparse weather stations that collect hourly or higher frequency data. Thus it is impossible to obtain sub-daily resolution by direct interpolation across weather stations. The current available hourly meteorological forcing is usually disaggregated from coarse temporal resolution. For example, the hourly NLDAS is disaggregated from NARR 3-hourly frequency. Previous studies have shown that NLDAS had large discrepancies towards SWE in higher elevation where lower SWE was simulated (Sheffield et al., 2003; Maxwell and Condon,

2016). Air temperature has also been shown to be systematically colder in winter and warmer in the spring months compared to the observations (Pan et al., 2003). These biases could be attributed to the $\sim 12\ km$ spatial resolution that greatly smoothed the local topographic variability.

### 4.3   Limitations, implications, and transferability of current study

There is a lack of high-resolution observation data to compare with the simulated variables. For example, the snow survey

from ASO has only been conducted a total of four times at this watershed and misses the temporal dynamics of snow depth. There is also not a single SNOTEL station within the watershed that we can use to compare simulated SWE at point location with the observed SWE. In addition to snow, we also do not have high-resolution ET data. Although MODIS provides an 8-day composite ET, it is relatively coarse compared to the temporal resolution used in the study. In the subsurface, there is no observed groundwater table depth or soil moisture data that can be used for the comparison. The remote sensed soil moisture

product (9 to 36 km resolution) from Soil Moisture Active Passive (SMAP) is likely to be too coarse to have any meaningful comparison.

       Uncertainty in the meteorological forcing has not been quantified. There is undoubtedly uncertainty in each GMF that may impact the simulated watershed responses, however, this is not the focus of this study. Precipitation collected from ground-based gages often has measurement uncertainty, which could lead to large uncertainty in interpolated gridded data sets especially in

the mountainous region where the gage network is sparse (Schreiner-McGraw and Ajami, 2020). In general, PRISM is assumed to perform better in matching gage observation in the mountainous regions compared to Daymet because of the relatively complex interpolation method (Daly et al., 2008). However, PRISM does not outperform Daymet in all gages (see Figure 3). Further, in areas where the terrain is flat and the gage network is dense, Daymet might perform better than, if not equally





well as, PRISM. Therefore, it is important to put the results into context when comparing different meteorological forcing in a
watershed setting.

Uncertainty in model parameterization has not been investigated, however, it does not change the conclusions of this study as
all simulations use the same set of model parameters except for meteorological forcing. It is well known that model parameters
such as subsurface structure and properties impact surface and subsurface flows and consequently ET and water storage. As
shown in this study, models using different meteorological forcing may produce dramatically different watershed responses
including streamflow. This has important implications on model calibration when the objective is to minimize the differences
between simulated and observed streamflow, which is true for most watershed hydrologic model calibration studies (Cromwell
et al., 2021). The choice of GMF affects the simulated streamflow, and in turn, the optimal parameters that are calibrated using
the simulated streamflow. Elsner et al. (2014) showed that there were substantial differences in calibrated model parameters
and simulated water balance using four different meteorological forcing for the same watershed. As a result, the choice of
meteorological forcing plays a critical role in model calibration and thus long-term planning and watershed management using
such calibrated model.

In this study, we choose Coal Creek as an illustrative example to show the effects of meteorological forcing spatiotemporal
resolution on watershed simulations. The performance of different meteorological forcing on matching watershed responses
may depend on the case study watersheds, however, the findings of meteorological forcing spatiotemporal effects on watershed
responses could be transferable to other mountainous watersheds that are dominated by snow.

## 5    Conclusions

This study aimed to compare three widely available GMF (Daymet, PRISM, and NLDAS) and evaluate the impacts of spa-
tiotemporal resolution of meteorological forcing on simulated streamflow, ET, SWE, soil moisture, surface ponded depth, and
groundwater table in a snow-dominated mountainous watershed. The different spatial and temporal resolutions were generated
by either downscaling or upscaling the native meteorological forcing resolution. The resulting meteorological forcing was then
applied as input to drive the fully-distributed, integrated watershed model (ATS).

To evaluate the performance of meteorological forcing, one should compare all aspects of watershed hydrologic responses.
Daymet has the best match in simulated streamflow, however, the simulated spatially distributed SWE has a weaker correlation
with the observed ASO SWE compared to that from PRISM. NLDAS performs the worst in both simulated streamflow and
spatially distributed SWE due to its coarsest grid resolution. Overall, NLDAS provides the most comprehensive dataset with
the highest temporal resolution (hourly) but comes with a spatial resolution of $\sim 12\ km$ that is too coarse for watershed
simulation, especially areas with complex terrain. On the contrary, both PRISM (800 m) and Daymet (1 km) provide finer
spatial resolution, capable of simulating watershed hydrological variables at high resolution, though they do not have the
complete forcing datasets. Using precipitation and temperature from PRISM in combination with solar input from Daymet
provides an alternative to drive watershed simulation with relatively high accuracy.



Using different spatial resolutions of PRISM ranging from 400 m to 4 km, the simulated discharge shows a minor difference when spatial resolution is $< 4\ km$ (or the grid area is $< 30\%$ of the watershed area). Similarly, the watershed average SWE, ET, total water storage do not show a significant difference between different spatial resolutions. Spatial resolution becomes more important when simulating spatially distributed hydrologic variables such as SWE and groundwater table.

Using a different temporal resolution of NLDAS (hourly to daily), the simulated discharge showed better performance with daily resolution compared to that forced by 12-hourly and hourly resolution. However, models forced by the sub-daily resolution preserve the dynamic watershed responses (e.g., diurnal fluctuation of streamflow) that are absent in results forced by daily resolution. This may have important implications on watershed biogeochemical reactions that often happen at sub-daily time scales.

It is difficult to choose the "best" meteorological forcing dataset because each dataset has its strengths and weaknesses, and what is best depends on the quantity of interest and its spatial and temporal scale. Ideally, we want to choose the GMF with the finest spatiotemporal resolution while providing the most complete forcing. None of the three GMFs are perfect. An alternative approach is to use meteorological forcing outputted from climate models such as WRF, which has the flexibility of generating much finer spatial and temporal resolution output while providing all available meteorological forcing datasets.

The choice of GMF affects the simulated streamflow and thus has an important implication on model calibration when the objective is to minimize the differences between simulated and observed streamflow. The findings of the effects of meteorological forcing spatiotemporal resolution on watershed simulations could be transferable to other mountainous watersheds that are snow-dominated.

*Code and data availability.* The datasets and scripts used in this study can be found on ESS-Dive: https://doi.org/xxxx. The downscaled
PRISM/Daymet at 400 m resolution is available at https://data.ess-dive.lbl.gov/view/doi:10.15485/1822259.

## Appendix A

### A1    Calendar day definition used in meteorological datasets

**Table A1.** Calendar day as defined in meteorological datasets and USGS gage

| Meteorological forcing | Calendar day in local timezone [UTC-05] | Calendar day in UTC |
| --- | --- | --- |
| PRISM | 07:00 $D^{-1}$ to 06:59 $D$ | 12:00 $D^{-1}$ to 11:59 $D$ |
| Daymet v4 | 07:00 $D$ to 06:59 $D^{+1}$ | 12:00 $D$ to 11:59 $D^{+1}$ |
| NLDAS-2 | 19:00 $D^{-1}$ to 18:59 $D$ | 00:00 $D$ to 23:59 $D^{+1}$ |
| USGS gage | 00:00 $D$ to 23:59 $D$ | 05:00 $D$ to 04:59 $D^{+1}$ |

Abbreviations: $D^{-1}$: previous day; $D$: current day; $D^{+1}$: next day.





## A2  Comparison of model outputs from different spatial resolution of Daymet

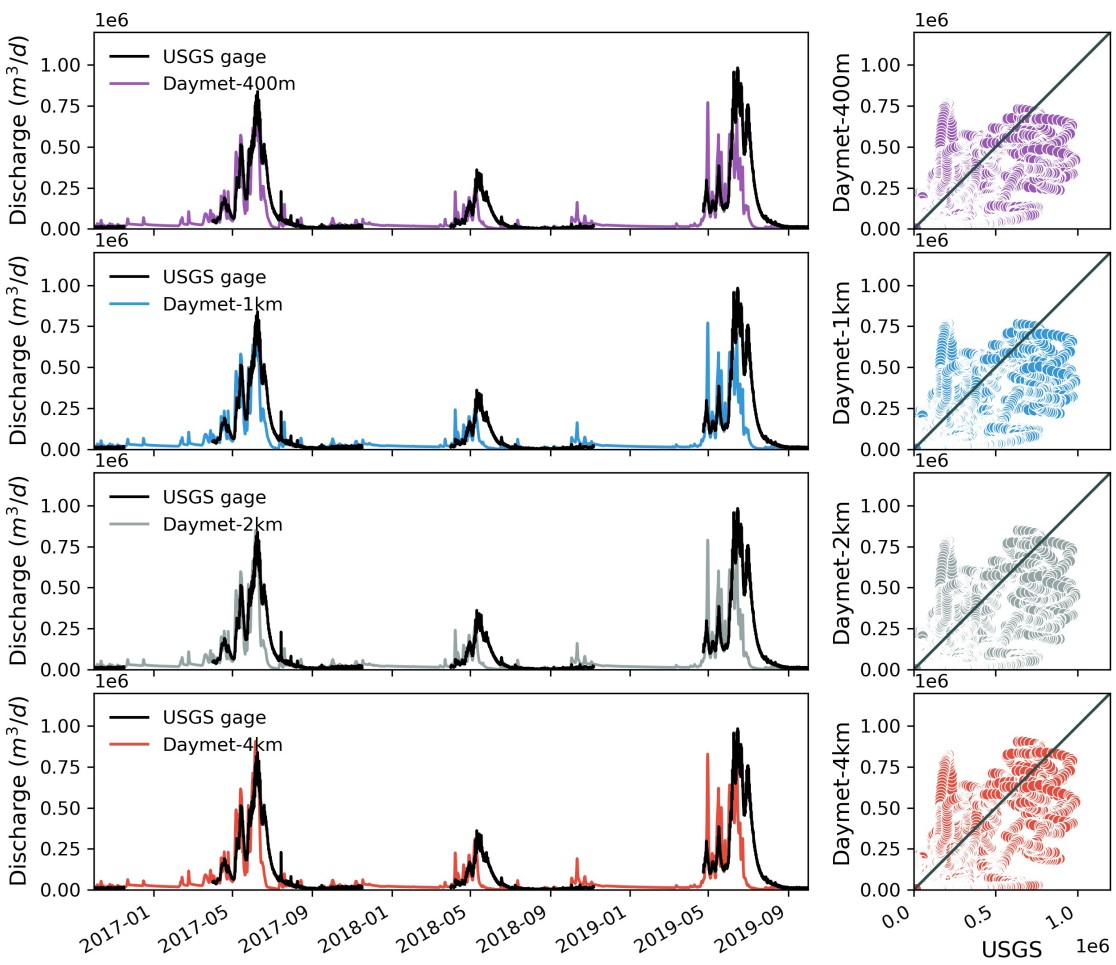

**Figure A1.** Simulated discharge at watershed outlet compared with USGS gage under different spatial resolution of Daymet. Also shown is the one to one plot.

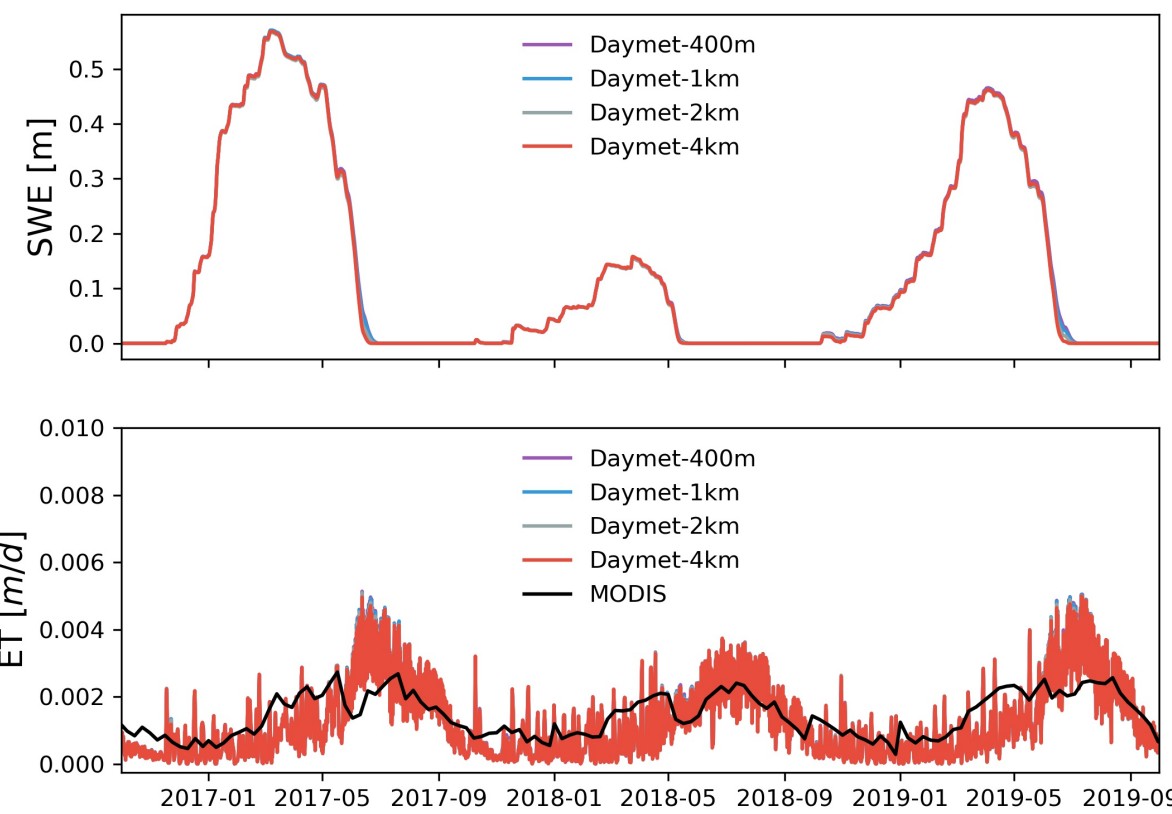

**Figure A2.** Simulated watershed average SWE and ET under different spatial resolution of Daymet.





## A3 Additional results from PRISM comparison under different spatial resolution

**Figure A3.** Spatial distribution of precipitation under different spatial resolution of PRISM at April 27, 2019.



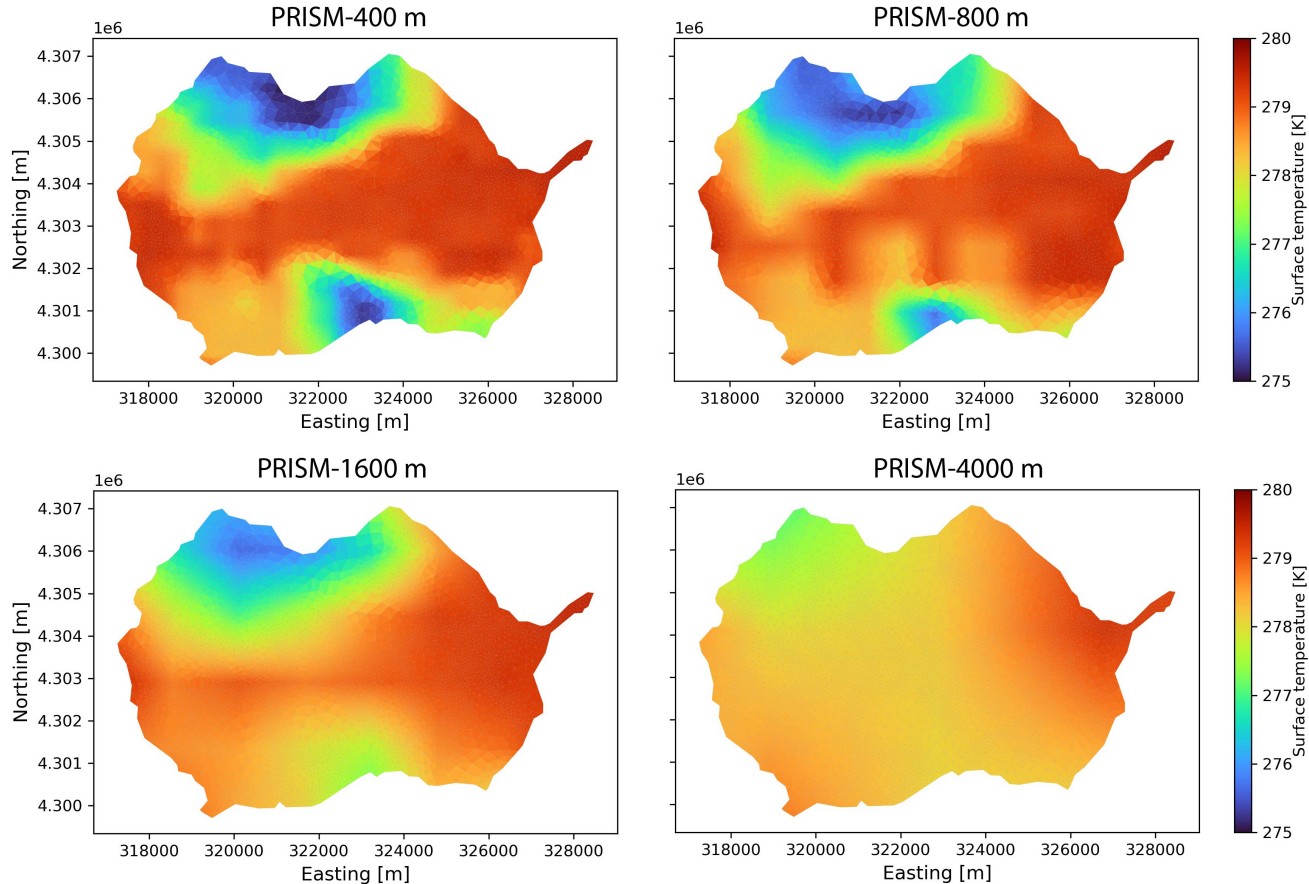

**Figure A4.** Spatial distribution of air temperature under different spatial resolution of PRISM at April 27, 2019.





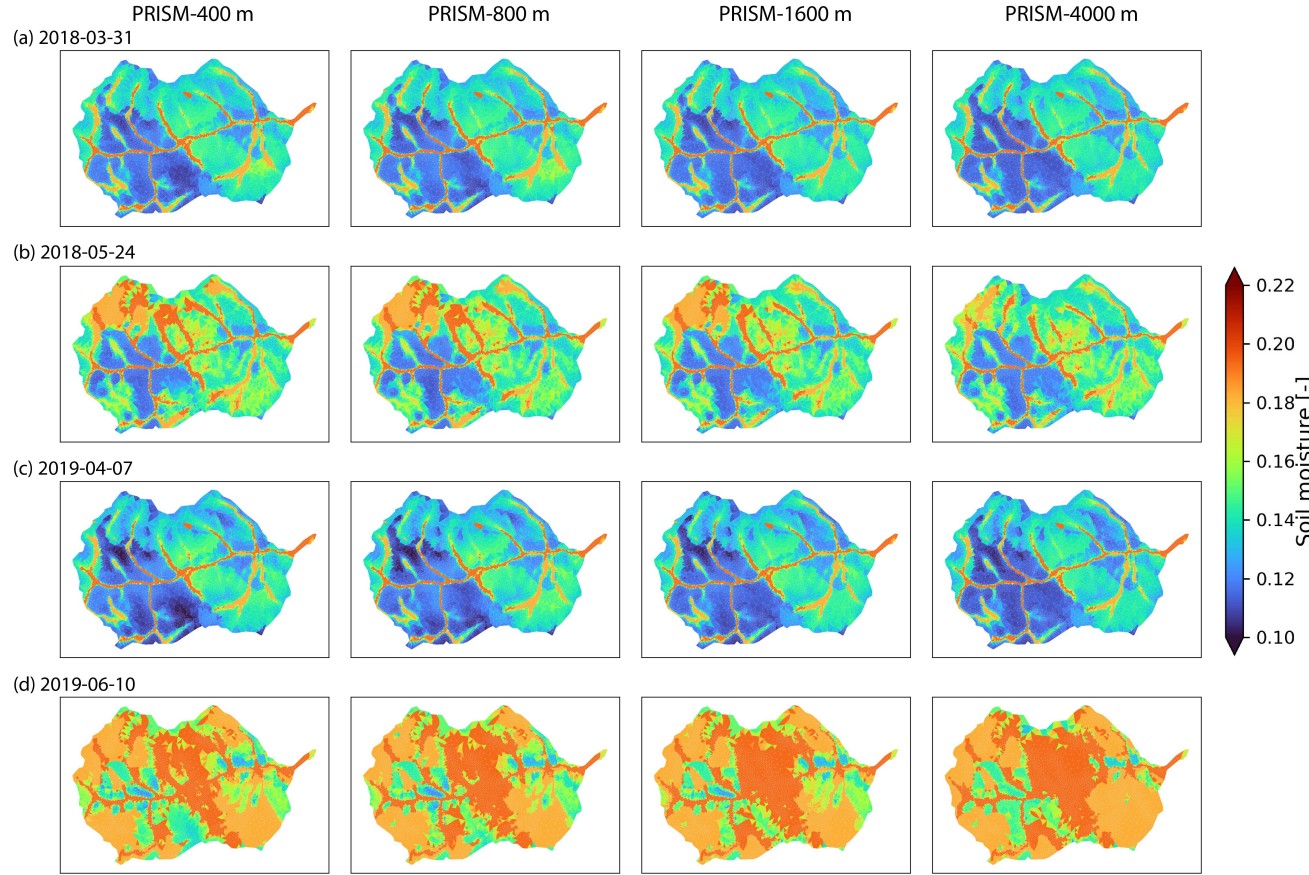

**Figure A5.** Spatial distribution of soil moisture under different spatial resolution of PRISM at four different times.





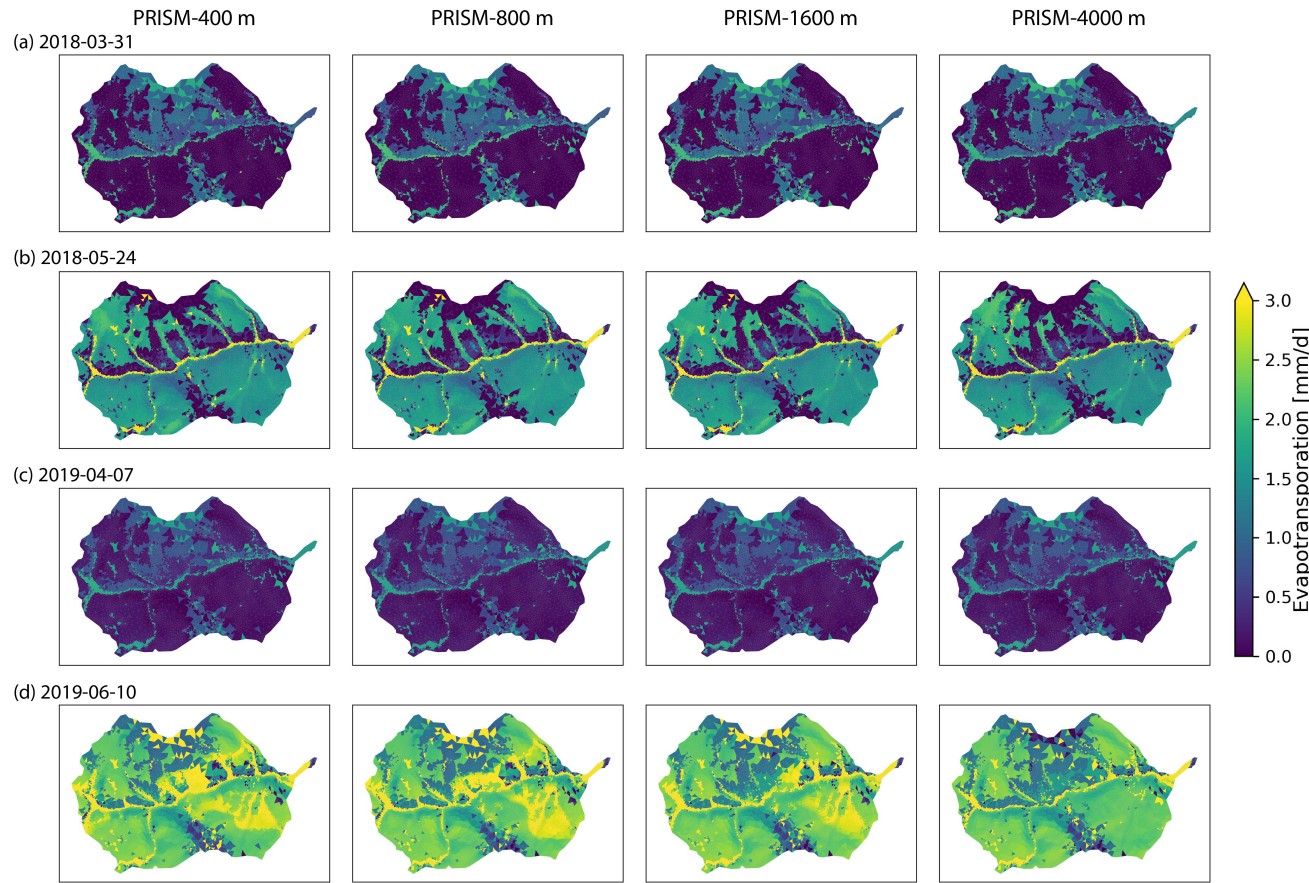

**Figure A6.** Spatial distribution of ET under different spatial resolution of PRISM at four different times.





*Author contributions.* PS designed the numerical experiments and performed the simulations with inputs from XC. UM and DD provided the PRISM 800-m product and UM performed the downscaling of meteorological forcing using machine learning. PS prepared the manuscript with contributions from all co-authors. All authors provided critical feedback and inputs to the manuscript.

*Competing interests.* The authors declare that they have no conflict of interest.

*Acknowledgements.* This research used resources of the National Energy Research Scientific Computing Center (NERSC), a DOE Of-
fice of Science User Facility supported by the Office of Science of the United States Department of Energy under contract DE-AC02-05CH11231. The proprietary PRISM data (800 m resolution) was purchased with funding from the Watershed Function Scientific Focus Area funded by the U.S. Department of Energy, Office of Science, Office of Biological and Environmental Research under Award no. DE-AC02-05CH11231. Pacific Northwest National Laboratory is operated for the DOE by Battelle Memorial Institute under contract DE-AC05-76RL01830. Oak Ridge National Laboratory is managed by UT- Battelle, LLC for the U.S. Department of Energy under Contract Number
DE-AC05-00OR22725. This paper describes objective technical results and analysis. Any subjective views or opinions that might be ex-pressed in the paper do not necessarily represent the views of the United States Department of Energy or the United States Government. This manuscript has been co-authored by UT- Battelle, LLC under Contract No. DE-AC05-00OR22725 with the U.S. Department of Energy. The United States Government retains and the publisher, by accepting the article for publication, acknowledges that the United States Government retains a non- exclusive, paidup, irrevocable, world-wide license to publish, or reproduce the published form of this manuscript, or allow
others to do so, for United States Government purposes. The Department of Energy will provide public access to these results of federally sponsored research in accordance with the DOE Public Access Plan (http://energy.gov/downloads/doe-public-access-plan).



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
