# Peer review of "The Effects of Spatial and Temporal Resolution of Gridded Meteorological Forcing on Watershed Hydrological Responses"

_Hydrology and Earth System Sciences, 2021_

## Author Comment (AC1)

**Response to RC 1**

**General Remarks:** The paper explores the impacts of publicly available gridded meteorological forcing (GMFs) on the simulated hydrology using an integrated hydrologic model. The choices of the GMFs are excellent as they are widely used in the hydrology community of the continental US. The selected site also allows analyzing different and complex settings (different topography, geology, and land cover) and is characterized by a heterogeneous pattern of forcing. The model is very suitable for this kind of study. Overall, the manuscript is well written and the objectives as well as the methodology are adequate and clearly explained.

**Response**: We thank the reviewer for the thoughtful and constructive feedbacks. The comments and suggestions are very helpful for improving this manuscript. We have addressed all the comments/concerns point-by-point below (highlighted in blue).

**Major Comments**

**Reviewer Comment 1.1** — Although the authors acknowledge the mismatch between the coarse resolution of the NLDAS and the size of their watershed, they didn't elaborate more on the impacts of this match. The watershed size is half the NLDAS grid and the precipitation is homogeneous, so technically NLDAS isn't adequate for this particular watershed, and analyzing the impacts of temporal resolution using NLDAS isn't accurate. The authors could analyze the impacts of the temporal resolution by using PRISM data instead of NLDAS. PRISM data could be converted from daily to hourly using NLDAS patterns for example.

**Response**: After consideration, we agree with the reviewers' concerns that the spatial scale of NLDAS is inappropriate for the temporal resolution study. To improve the manuscript, we chose to use temporally downscaled PRISM data to study the effects of temporal resolution of meteorological forcing on watershed responses. Not surprisingly, the conclusions are still the same, although the metrics are slightly different. The simulated streamflow shows better performance using daily resolution of PRISM compared to that using sub-daily resolution of PRISM. We have updated both the comparison plots and the metric table in the revised manuscript.

**Reviewer Comment 1.2** — The authors compare both the impacts of spatial and temporal resolution, but they did so using different forcing. Hence, it is difficult to highlight which one has the highest impact on the simulated hydrology. By analyzing the impacts of the temporal resolution using PRISM, the authors could compare the impacts of spatial and temporal resolution and highlight the most important resolution for this particular watershed.

**Response**: Please see the above response, which discussed about choosing PRISM for both spatial and temporal resolution study.

**Reviewer Comment 1.3** — What are the physical explanations for the differences between NLDAS, PRISM, and DAYMET? Are these differences due to the physics, or the types of data they used?

**Response**: The differences in the met forcing can be attributed to both the data source and the interpolation method used in each dataset (see Table 1). Daymet primarily uses GHCN-D for precipitation and temperature observations (Thornton et al., 2021). PRISM uses GHCN-D in addition to a number of local/state weather stations (see Table A1 in https://prism.oregonstate.edu/documents/PRISM_datasets.pdf). On the contrary, NLDAS primarily uses model reanalysis data while adjusting for observations (https://hydro1.gesdisc.eosdis.nasa.gov/data/NLDAS/README.NLDAS2.pdf). There have been a number of studies looking at the differences between NLDAS, PRISM and Daymet in different areas (e.g., Behnke et al., 2016). They found that different interpolation methods affected the accuracy of downscaled meteorological data. It is beyond the scope of this study to provide a detailed comparison between these met forcing dataset. However, we have updated Table 1 to provide more detailed information about each dataset. We have also added a few sentences in Section 2.3 to provide more details.

**Reviewer Comment 1.4** — It is difficult to compare data with different spatial and temporal resolutions in addition to the differences in the methods used to generate these datasets. How can we differentiate errors due to the spatial and temporal resolution of the GMFs from the methods used to generate them?

**Response**: The downscaling and upscaling approaches of met forcing are inherently different because downscaling requires additional information for disaggregation, but aggregating available data can achieve upscaling. For example, we used Random Forest for spatial downscaling (from 800-m to 400-m) and moving average for spatial upscaling (from 800-m to 1600-m or 4000-m). Although the approaches are different, we ensure that the aggregated total precipitation and mean temperature stay the same across different spatial resolutions. Even without the 400-m resolution comparison, the conclusions of the spatial resolution effects on watershed responses would still hold. For temporal upscaling, we used the same approach for aggregating hourly-PRISM into 12-hourly and daily dataset. We also ensure that the aggregated total precipitation and mean temperature stay the same across different temporal resolutions.

**Reviewer Comment 1.5** — Because these datasets provide different variables, it could be great to know which of these forcing variables (e.g., precipitation or temperature) drive the observed impacts.

Is a high-resolution precipitation more important that a high-resolution temperature or solar radiation?

**Response**: We agree that this is an important question to address and will greatly improve our understanding of the importance of different forcing variables. In our revision, we systematically varied each variable one at a time to isolate the effect of individual forcing. We now compared Daymet, PRISM and NLDAS by using their precipitation, or temperature or both in the model while keep the other variables the same. Not surprisingly, the resolution of precipitation plays a more important role than the resolution of temperature in streamflow simulations. Because PRISM does not provide solar radiation, we did not compare solar radiation across different GMFs. Instead, we tested the sensitivity of solar radiation spatial resolution using Daymet and temporal resolution using NLDAS. We found that the spatial resolution of shortwave radiation has little impact on ET and streamflow, whereas the temporal resolution of shortwave radiation slightly changed the dynamics of ET and streamflow.

**Reviewer Comment 1.6** — The authors used the DAYMET solar radiation when running the model with the PRISM dataset, as their results have shown by merging these datasets some

simulated variables such as ET were found to be equal. What are the impacts of solar radiation? Is a high-resolution solar radiation needed to simulate the system? The energy balance (ET and SWE) plays an important role in this watershed, and because most of the key forcing variables were the same, so were the simulated hydrologic variables.

**Response**: We agree that solar radiation may play an important role in the watershed. To address the reviewer's questions, we tested the effect of solar radiation spatial resolution using Daymet shortwave radiation at 1km, 2km and 4km resolution while keeping the other forcings the same. We then tested the effect of solar radiation temporal resolution using NLDAS shortwave radiation at hourly, 12-hourly and daily resolution while keeping other forcings the same. We found that the spatial resolution of shortwave radiation has little impact on ET and streamflow, whereas the temporal resolution of shortwave radiation slightly changed the dynamics of ET and streamflow. Overall watershed responses are less sensitive to solar radiation than precipitation and temperature. We have included the new results in the revised manuscript.

**Reviewer Comment 1.7** — The authors should isolate the impacts of the different variables. They could perform a simulation with different precipitation (PRISM, NLDAS, DAYMET) and keep the other variables identical, then the same temperature and precipitation and the other variables remaining identical. This will allow to better understand how the uncertainties of these variables drive the observed differences. Performing the comparisons step by step will allow providing insights into how each components acts and the aggregated effects.

**Response**: Please see the response to Reviewer Comment 1.5.

**Reviewer Comment 1.8** — State from the beginning that there are no ground measurements of groundwater levels and soil moisture. This is important to better understand the goal of the study.

**Response**: We have added the following sentence in Section 2.4 under Observation Data.

*Groundwater measurements and field observed soil moisture data are not available within the study site.*

**Reviewer Comment 1.9** — In the absence of a reasonable number of ground measurements, the authors could compare the random errors embedded in each of the datasets. The triple collocation analysis allows computing the random errors associated with each dataset without knowing the truth (ground measurements). The authors could employ the framework and better analyze the differences between the GMFs first and the simulated hydrology. Otherwise, it is hard to know which dataset is accurate or contains fewer biases.

**Response**: Thank you for the suggestion. We have performed Triple Collocation Analysis (TCA) for precipitation and temperature using the three GMFs for the larger East-Taylor watershed. We also compared TCA results at the GHCN-D gauge locations with the results from the Taylor Diagram analysis we have performed under Section 3.1. The results are quite similar. For both precipitation and temperature, Daymet and PRISM have less error variance compared to NLDAS. We will put the TCA results in the appendix.

**Reviewer Comment 1.10** — Are the findings watershed-dependent? How can we apply these findings to other watersheds?

**Response**: Our study area–Coal Creek is one of the many mountain headwater catchments within the Upper Colorado River Basin, which is also one of the principal headwater basins in the U.S. and provides water for over 40 million people. The watershed is snow-dominated with strong variations in topography and land cover, which is an ideal site for testing heterogeneous spatial and temporal pattern of meteorological forcing. Our conclusions would hold for other mountainous headwater watersheds that are dominated by snow, and additional studies are needed to evaluate the GMF in other areas that are not dominated by snow. In addition, performing such studies in many watersheds is computationally expensive. We realize that this is a limitation and we have expanded the discussion on the transferability of current study in the revised manuscript.

*In this study, we choose Coal Creek as an illustrative example to show the effects of meteorological forcing spatiotemporal resolution on watershed simulations. The study site has strong variations in topography and land cover, which is an ideal site for testing heterogeneous spatial and temporal pattern of meteorological forcing. Our conclusions would hold for other mountainous headwater watersheds that are dominated by snow because we did not make any site-specific assumptions. However, additional studies are needed to evaluate the GMF in other areas that are not dominated by snow.*

**Minor Comments**

**Reviewer Comment 1.11** — WRF-Hydro is not an integrated hydrologic model

**Response**: WRF-Hydro has been removed from the list of integrated hydrologic models.

**Reviewer Comment 1.12** — Paragraph 100: add the source of the annual precipitation values

**Response**: The precipitation source is added.

*It receives 850 mm of precipitation annually, with 530 mm as snowfall which was estimated from long-term Daymet forcing dataset (Thornton et al., 2021)*

**Reviewer Comment 1.13** — Does the water table depth (in areas located outside the watershed because there are no measurements in the watershed) go beyond 28m depth? what are the boundary conditions at the lower limits and the other limits?

**Response**: The bottom of the model domain is determined from the global depth-to-bedrock dataset, which is the best available dataset for this region. Therefore, we choose the largest depth-to-bedrock as a confining layer and assume everything below 28-m is impermeable. We set model boundary at the bottom and sides to no-flow. There is no available well data near this watershed that we can use to validate our assumption.

**Reviewer Comment 1.14** — What do you mean by high snow year and low snow year, please provide numbers

**Response**: We have added the total snow precipitation for high/low snow year.

*The study period features a high snow year ( 709 mm in water year 2017) and a low snow year ( 296 mm in water year 2018)...*

**Reviewer Comment 1.15** — How did you test the pinup. Did you compare the storages? How did you find that the spin-up period is accurate?

**Response**: For the 10-year spinup (2004-2014), we checked the accumulative total water storage changes over time and it was in dynamic equilibrium (i.e., the water storage gained and lost in response to flow events). We also discarded the first year run from all the simulations to ensure that the results were not affect by initial conditions.

**Reviewer Comment 1.16** — Even if the model is not calibrated it should be able to reproduce the measured streamflow?

**Response**: ATS is a process-based, fully distributed model based on physical equations. Even without calibration, the model was able to perform reasonably well (KGE=∼0.6 using Daymet) in streamflow. We have tested ATS in other watersheds and have seen similar encouraging results (manuscript is under preparation, see AGU abstract here: https://agu.confex.com/agu/fm21/meetingapp.cgi/Paper/899787).

**Reviewer Comment 1.17** — Figure 1: Add the location of the watershed in the US or at least the western US. What is NHDPlus, add the definition and the source of the dataset

**Response**: We have added a map showing the location of the study area within the state of Colorado. We have also added the definition and source of NHDPlus in the figure caption.

**Reviewer Comment 1.18** — Figure 2: missing legend of soil and geology map

**Response**: We have intentionally left out the soil/geology map legend due to too many colorbar indexes. We think the current plot is sufficient to show the heterogeneous distribution of soil and geology without revealing too much details.

**Reviewer Comment 1.19** — Figure 4: put the temporal variations figures on the same graph for visual comparisons

**Response**: We have plotted all time series discharge plots in one figure. To allow better visibility, we also plotted the flow duration curves.

**Reviewer Comment 1.20** — Figure 6: smoothen ET for a better comparison

**Response**: In the middle plot, we would like to show the differences in temporal dynamics between different forcing. For a better comparison, we plotted their 8-day accumulative values in the bottom plot.

**Reviewer Comment 1.21** — Figure 7: Plot the differences

**Response**: We have plotted the SWE spatial difference between model simulations and observation. However, the magnitude of difference was too large to allow a better comparison. The simulated SWE were very small compared to observed SWE. In some places, the differences could reach 2m. Here we are more interested in the spatial correlation of SWE instead of the absolute differences.

**Reviewer Comment 1.22** — Figure 6: ET is misleading. Compare daily and hourly ET similarly with the same units.

**Response**: Sorry for the confusion. Both ET driven by daily and hourly GMF were plotted at hourly timestep with the same unit (i.e., mm/d). The ET driven by hourly GMF showed more dynamic fluctuations, which is expected.

**Reviewer Comment 1.23** — Figure 8: plot the temporal variations figures together

**Response**: We have plotted all time series discharge plots in one figure. To allow better visibility, we also plotted the flow duration curves.

**Reviewer Comment 1.24** — Figure 10: plot the differences

**Response**: We have plotted the SWE spatial difference between model simulations and observation. However, the magnitude of difference was too large to allow a better comparison. The simulated SWE were very small compared to observed SWE. In some places, the differences could reach 2m. Here we are more interested in the spatial correlation of SWE instead of the absolute differences.

**Reviewer Comment 1.25** — Figure 12: Is the groundwater table equal to 3000m?

**Response**: Yes, the groundwater table is in absolute elevation (i.e., use sea level as datum). We also added the surface elevation at each location to show better comparison.

**Reviewer Comment 1.26** — Figure 13: plot the temporal variations figures together

**Response**: We have plotted all time series discharge plots in one figure. To allow better visibility, we also plotted the flow duration curves.

**Reviewer Comment 1.27** — Figure 15: Are the daily NLDAS ET is twice higher than the hourly NLDAS ET? Why the differences are less pronounced for the 8d?

**Response**: Sorry for the confusion. All ET were shown at hourly timestep with the same unit (i.e., mm/d). The difference in the dynamic pattern is caused by the temporal resolution of NLDAS forcing. Hourly NLDAS forcing induced more dynamic pattern compared to 12-hourly and daily. If we convert the hourly timestep to daily, the pattern would be very similar. The 8-d ET is a cumulative value calculated by summing ET within 8-d time window, so the cumulative values were about the same.

**Reviewer Comment 1.28** — What explains the low snow associated with hourly NLDAS? Is the temperature higher with daily NLDAS?

**Response**: Were you referring to Figure 15? The SWE (as a proxy for snow depth) is impacted by snowmelt rate, which is further determined by temperature and solar radiation. Hourly NLDAS

induced slightly higher overall ET than those from 12-hourly and daily NLDAS (see 8-d ET in Figure 15). Although the average daily air temperature is the same for all NLDAS temporal resolution, air temperature forced by hourly-NLDAS is more dynamic than air temperature forced by daily-NLDAS. It was clear from the snowmelt rate plot in Figure 14 that hourly NLDAS showed larger variations in snowmelt rate than that from daily NLDAS. Therefore, the low snow associated with hourly NLDAS is caused by the combined effect of fast snowmelt and large ET.

**Reviewer Comment 1.29** — Paragraph 360 is cut in my version "Additionally?"

**Response**: The extra word has been deleted.

**References**

---

## Author Comment (AC2)

**Response to RC 2**

**General Remarks:** This paper investigates the impacts of meteorological forcing in simulating rainfall-runoff behavior using the ATS model. Three different datasets are used to test the effects of spatial and temporal resolution of gridded meteorological forcing on surface runoff and other related hydrological processes. This is an interesting piece of work that provides certain decision support for the application of meteorological forcing.

**Response**: We thank the reviewer for the thoughtful and constructive feedbacks. The comments and suggestions are very helpful for improving this manuscript. We have addressed all the comments/concerns point-by-point below (highlighted in blue).

**Major Comments**

**Reviewer Comment 2.1** — This study only tested on the Coal Creek Watershed, which has a relatively small size of about 53km2 and receives a high proportion of snowfall during cold season. Would the results be different if the gridded meteorological forcing was tested in other catchments? Is the result for individual catchment representative of general trends?

**Response**: We thank the reviewer for the critical comments. This question has also been raised by Reviewer 1. Please refer to the response in RC 1.10. For your convenience, we have pasted our response here:

Our study area–Coal Creek is one of the many mountain headwater catchments within the Upper Colorado River Basin, which is also one of the principal headwater basins in the U.S. and provides water for over 40 million people. The watershed is snow-dominated with strong variations in topography and land cover, which is an ideal site for testing heterogeneous spatial and temporal pattern of meteorological forcing. Our conclusions would hold for other mountainous headwater watersheds that are dominated by snow, and additional studies are needed to evaluate the GMF in other areas that are not dominated by snow. In addition, performing such studies in many watersheds is computationally expensive. We realize that this is a limitation and we have expanded the discussion on the transferability of current study in the revised manuscript.

*In this study, we choose Coal Creek as an illustrative example to show the effects of meteorological forcing spatiotemporal resolution on watershed simulations. The study site has strong variations in topography and land cover, which is an ideal site for testing heterogeneous spatial and temporal pattern of meteorological forcing. Our conclusions would hold for other mountainous headwater watersheds that are dominated by snow because we did not make any site-specific assumptions. However, additional studies are needed to evaluate the GMF in other areas that are not dominated by snow.*

**Reviewer Comment 2.2** — The performance criterion is critical for model application. This study took modified Kling-Gupta efficiency as objective function to evaluate the model performance, which means that you specially focus on water balance. The results can be very sensitive to the selection of objective function. Have you considered some other performance criteria? Do they show different comparisons?

**Response**:   We thank the reviewer for the suggestion of considering other metrics. In fact, we have used other metrics including the original KGE [Gupta et al., 2009], Nash-Sutcliffe Efficiency (NSE), and log(NSE) for evaluating the model performance during manuscript preparation. However, each metric focuses on a specific part of the model performance and has its own pitfalls [Clark et al., 2021]. For example, NSE is highly influenced by high flows whereas log(NSE) places a large emphasis on low flows. The modified KGE avoids the effect of input bias on the variability indicator which has an advantage over the original KGE [Kling et al., 2012]. Another advantage of using modified KGE is that it can be decomposed into three different constitutive parts (i.e., correlation coefficient, variability and bias) and thus could be used to diagnose the model performance score. In general, modified KGE follows the same trend of NSE. As a result, we choose to use a single metric that is physically explainable for better comparison of model performance. However, we can add the other metrics in the manuscript if the reviewer strongly encourage to do so.

**Reviewer Comment 2.3** — The authors say that "The models were not calibrated because the focus of this study was to evaluate the effect of meteorological forcings on model simulation instead of estimating the optimal parameters used in ATS. " The simulation results may vary for different parameters, and the parameters calibrated for different spatial and temporal resolutions may also be different. It can be seem from the hydrographs that the simulated discharge differ significantly from the observation and are underestimated for most of the time period. Does it have any effect on the results if the model parameters are calibrated in advance?

**Response**:   We agree that the simulation results may vary if we use calibrated parameters, and the calibrated parameters may be different depending on the spatial and temporal resolution of GMF and the targeted hydrologic variables used during calibration. Therefore, we think that it is not appropriate to use the calibrated parameters from one specific spatial and temporal resolution of GMF to test the impact on model simulations forced by different spatial and temporal resolutions of GMFs. Further, the focus of the study was not on model calibration, but rather an evaluation of the effects of different GMFs on simulated watershed responses. We think that adding model calibration would detract from the current emphasis on GMF spatiotemporal effects. Although the current model is not calibrated, it still performs reasonably well (KGE=∼0.6 using Daymet) for streamflow.
In fact, we are working on a separate paper that uses novel machine learning technique for estimating model parameters for this watershed. In that paper, we are also investigating the impact of using different GMF on model calibration. We hope to cite that work when it becomes available.

**Reviewer Comment 2.4** — The authors used different meteorological forcing to compare the effect of different temporal and spatial resolutions on the simulation results, which makes it difficult to state which is the greater impact to the simulation results. It is recommended that improvements in model performance at different temporal and spatial resolutions be investigated on the basis of the same data sources.

**Response**:   We thank the reviewer for the excellent suggestion. Similar comments have also posted by Reviewer 1 (see RC1 comments 1.1 and 1.2). For your convenience, we have pasted our response here:

To be consistent with the spatial resolution comparison, we now use temporally downscaled PRISM data to study the effects of temporal resolution of meteorological forcing on watershed responses. Not surprisingly, the conclusions are still the same, although the metrics are slightly different. The simulated

streamflow shows better performance using daily resolution of PRISM compared to that using sub-daily resolution of PRISM. We have updated both the comparison plots and the metric table in the revised manuscript. Also, we systematically varied each variable one at a time to isolate the effect of individual forcing. We now compared Daymet, PRISM and NLDAS by using their precipitation, or temperature or both in the model while keep the other variables the same. Not surprisingly, the resolution of precipitation plays a more important role than the resolution of temperature in streamflow simulations

**Specifc Comments**

**Reviewer Comment 2.5** — The catchment is very small, what is the average time of concentration for the floods? Does it have an impact on the comparison of daily flood simulation results?

**Response**: To clarify, all the discharge plots were using hourly data. So we were comparing hourly simulated discharge with hourly observed discharge. Because this catchment is snow-dominated, most of the precipitation would first infiltrate into the subsurface as groundwater and then exfiltrate out at the stream channels. Although there is no direct way to estimate the average time of concentration, it is less likely to be a very short time ($< 1$ hour). In addition, ATS was taking sub-hourly timestep and the simulate discharge was integrated over hourly time window.

**Reviewer Comment 2.6** — The grid size of NLDAS is larger than the actual area of the basin, it doesn't seem appropriate as a grid input.

**Response**: As has discussed in the manuscript, the coarse resolution of NLDAS would make the forcing highly uniform over a small catchment. As a result, NLDAS is commonly used in large basin-scale models. However, because NLDAS has the most complete dataset with hourly temporal resolution, it is still an attractive product for watershed model simulations. In this study, we were evaluating the performance of NLDAS for a small watershed to see if NLDAS could still provide any value. We found that NLDAS performed poorly for streamflow and other variables, but it preserved the dynamic watershed responses which were critical if interested in simulating flash flood or hyporheic exchange.

**Reviewer Comment 2.7** — For Figure 4(also Figure 8 and 13), is it possible to show all the simulation results in one graph to have a better comparison of the different results?

**Response**: We have plotted all discharge time series in one graph and used flow duration curve to show better comparison.

**Reviewer Comment 2.8** — It is difficult to distinguish the differences between the results in Figure 11. Is it possible to have a better comparison by using flow duration curve?

**Response**: We thank the reviewer for the suggestion. However, flow duration curve is commonly used for discharge and we were comparing surface ponded depth in Figure 11. We will zoom into high flow period to show more details between different depths.

**References**

Martyn P. Clark, Richard M. Vogel, Jonathan R. Lamontagne, Naoki Mizukami, Wouter J.M. Knoben, Guoqiang Tang, Shervan Gharari, Jim E. Freer, Paul H. Whitfield, Kevin Shook, and Simon Papalexiou. The abuse of popular performance metrics in hydrologic modeling. *Water Resources Research*, aug 2021. ISSN 0043-1397. doi: 10.1029/2020WR029001. URL https://onlinelibrary.wiley.com/doi/10.1029/2020WR029001.

Hoshin V. Gupta, Harald Kling, Koray K. Yilmaz, and Guillermo F. Martinez. Decomposition of the mean squared error and NSE performance criteria: Implications for improving hydrological modelling. *Journal of Hydrology*, 377(1-2):80–91, 2009. ISSN 00221694. doi: 10.1016/j.jhydrol.2009.08.003. URL http://dx.doi.org/10.1016/j.jhydrol.2009.08.003.

Harald Kling, Martin Fuchs, and Maria Paulin. Runoff conditions in the upper Danube basin under an ensemble of climate change scenarios. *Journal of Hydrology*, 424-425:264–277, 2012. ISSN 00221694. doi: 10.1016/j.jhydrol.2012.01.011. URL http://dx.doi.org/10.1016/j.jhydrol.2012.01.011.